# De novo genic mutations among a Chinese autism spectrum disorder cohort

Tianyun Wang[1,*], Hui Guo[1,2,*], Bo Xiong[3,*,†], Holly A.F. Stessman[3,*], Huidan Wu[1], Bradley P. Coe[3], Tychele N. Turner[3], Yanling Liu[1], Wenjing Zhao[1], Kendra Hoekzema[3], Laura Vives[3], Lu Xia[1], Meina Tang[1], Jianjun Ou[2], Biyuan Chen[4], Yidong Shen[2], Guanglei Xun[5], Min Long[1], Janice Lin[3], Zev N. Kronenberg[3], Yu Peng[1], Ting Bai[1], Honghui Li[6], Xiaoyan Ke[7], Zhengmao Hu[1], Jingping Zhao[2], Xiaobing Zou[4], Kun Xia[1,8,9] & Evan E. Eichler[3,10]

Recurrent de novo (DN) and likely gene-disruptive (LGD) mutations contribute significantly to autism spectrum disorders (ASDs) but have been primarily investigated in European cohorts. Here, we sequence 189 risk genes in 1,543 Chinese ASD probands (1,045 from trios). We report an 11-fold increase in the odds of DN LGD mutations compared with expectation under an exome-wide neutral model of mutation. In aggregate, ~4% of ASD patients carry a DN mutation in one of just 29 autism risk genes. The most prevalent gene for recurrent DN mutations is SCN2A (1.1% of patients) followed by CHD8, DSCAM, MECP2, POGZ, WDFY3 and ASH1L. We identify novel DN LGD recurrences (GIGYF2, MYT1L, CUL3, DOCK8 and ZNF292) and DN mutations in previous ASD candidates (ARHGAP32, NCOR1, PHIP, STXBP1, CDKL5 and SHANK1). Phenotypic follow-up confirms potential subtypes and highlights how large global cohorts might be leveraged to prove the pathogenic significance of individually rare mutations.

[1] The State Key Laboratory of Medical Genetics, School of Life Sciences, Central South University, Changsha, Hunan 410078, China. [2] Mental Health Institute, the Second Xiangya Hospital, Central South University, Changsha, Hunan 410011, China. [3] Department of Genome Sciences, University of Washington School of Medicine, Seattle, Washington 98195, USA. [4] Children's Development Behavior Center, Third Affiliated Hospital of Sun Yat-sen University, Guangzhou, Guangdong 510630, China. [5] Mental Health Center of Shandong Province, Jinan, Shandong 250014, China. [6] Child Healthcare Department, Liuzhou Maternity and Child Healthcare Hospital, Liuzhou, Guangxi 545000, China. [7] Child Mental Health Research Center, Nanjing Brain Hospital Affiliated of Nanjing Medical University, Nanjing, Jiangsu 210029, China. [8] Collaborative Innovation Center for Genetics and Development, Shanghai 200433, China. [9] Key Laboratory of Medical Information Research, Central South University, Changsha, Hunan 410013, China. [10] Howard Hughes Medical Institute, University of Washington, Seattle, Washington 98195, USA. * These authors contributed equally to this work. † Present address: Department of Forensic Medicine, Tongji Medical College, Huazhong University of Science and Technology, No. 13, Hangkong Road, Wuhan, Hubei 430030, China. Correspondence and requests for materials should be addressed to K.X. (email: xiakun@sklmg.edu.cn) or to E.E.E. (email: eee@gs.washington.edu).

Autism spectrum disorders (ASDs) are characterized by impairments in social communication and restricted or repetitive behaviours or interests. ASD has a strong but complex genetic component and is typified by a male bias[1]. Although the prevalence of ASD in the United States has increased to 1 in 68 children[2], the prevalence worldwide has been estimated as closer to 1% (ref. 3). Large-scale whole-genome copy number variation (CNV) studies[4–7], whole-exome sequencing studies[8–13] and candidate resequencing studies[14,15] have established the importance of de novo (DN) germline mutations in ASD. Studies of DN mutations have led to the discovery of dozens of new candidate ASD risk genes in primarily European cohorts. However, no study has explored on a large scale the spectrum of DN mutations in these ASD risk genes in the Chinese population. While the frequency of DN mutation should typically not differ among population groups, examples of ethnic predilection (for example, 17q21.31 recurrent mutations) due to genomic architecture, sequence composition or patient recruitment biases may exist[16–18].

In this study, we apply single-molecule molecular inversion probes (smMIPs)[19] to rapidly and cost-effectively sequence the coding regions of 189 autism risk genes across a large cohort of 1,543 autism probands (1,045 from trios) from the Autism Clinical and Genetic Resources in China (ACGC). Several candidate genes with recurrent DN likely gene-disruptive (LGD) mutations are established by this study, implicating several novel autism risk genes in the Chinese population. Wherever possible, we worked with the five ACGC coordinating centres to investigate patterns of inheritance and phenotypic features of patients identified with DN mutations. We compare these data to determine whether previously observed gene-specific comorbidities distinguish etiological subtypes of autism in the Chinese population. Our study helps to identify global risk genes providing evidence and motivation for further functional and translational studies of specific ASD risk genes, which may guide future personalized treatments.

## Results

**Resequencing of 189 genes in a large Chinese ASD cohort.** We selected autism risk genes for targeted sequencing mainly based on the frequency and severity of DN mutations from previously published exome sequencing studies[12,13] under the hypothesis that DN mutations in genes contributing to autism pathology will not differ significantly between global populations. These included DN mutation calls from exome sequencing of autism families—primarily the Simons Simplex Collection (SSC) and the Autism Sequencing Consortium (ASC). Candidate genes were ranked according to the presence of the following criteria: (1) recurrent DN LGD events, (2) one LGD event and more than one DN missense mutation, (3) recurrent DN missense mutations and CNV disruption among cases of ASD and developmental delay (DD)[3,5,6,20], (4) CNV disruption among cases of ASD and DD with potential functional relevance in the brain (yet no DN single-nucleotide variants in exome sequencing data from ASD probands) or (5) multiple DN events among intellectual disability (ID) probands[21,22]. We also chose to include several known ASD risk genes. We excluded genes that showed a high tolerance to severe mutations in American populations based on exome sequencing data from 6,500 unaffected individuals (NHLBI Exome Sequencing Project). The final gene set consisted of 213 candidate genes (Supplementary Data 1).

Molecular inversion probes (MIPs) were designed using an improved algorithm, which incorporates a single-molecule (sm) tag into the degenerate region within the MIP backbone[19]. The final designs for our 213 candidate genes included 11,394 smMIPs distributed into three pools after optimization (see the 'Methods'

section and Supplementary Data 2). For the purpose of this study, we assembled one of the largest Chinese ethnic cohorts of ASD. The ACGC aims to collect the clinical and genetic resources of more than 10,000 ASD families (trios/quads) over the next 5 years. All the patients in this study were born in China and were diagnosed according to DSM-IV (American Psychiatric Association, 2000) criteria (Supplementary Table 1). In total, 1,086 ASD proband–parent trios (one affected offspring and two unaffected parents) and 561 autistic children (most with one parent sample available) from the ACGC were selected for this study. The geographical distribution of the families is shown (Fig. 1 and Supplementary Table 2) based on the birth origin of the patients. Only probands ($n = 1,647$; male:female ratio $= 6$:1) were targeted for initial sequencing.

In total, 1,543 probands (1,045 from trios) passed MIP capture and other QC measures (see the 'Methods' section and Supplementary Fig. 1 and Supplementary Data 3) and 189 genes passed QC measures (see the 'Methods' section and Supplementary Fig. 2 and Supplementary Data 2). Further analysis of this data set was based only on those genes and samples that passed these QC measures. We discovered 4,226 rare variants (see the 'Methods' section) predicted to alter the amino acid sequence or gene splicing. We selected LGD (nonsense, frameshift and splice site) mutations ($n = 120$, of which 92 were found among the 1,045 trio probands) and missense mutations with a combined annotation dependent depletion (CADD) score of greater than 30 (MIS30; $n = 216$, of which 179 were found among the 1,045 trio probands) for validation. In total, we validated 321 putative severe mutations (115 LGD and 206 MIS30) by Sanger sequencing with an overall validation rate of 95.5%. Where parental DNA were available, we also assessed inheritance by Sanger sequencing of variants (Supplementary Data 4).

**Inheritance and mutation analysis.** Among the 1,045 trios, we discovered 43 DN mutations in 29 of the 189 candidate genes tested, of which 35 were LGD events (Supplementary Data 4). We calculated the overall probability of detecting 35 or more DN LGD events in our panel of 189 genes using a probabilistic model derived from human–chimpanzee differences and an expected rate of 1.75 DN mutations per exome[14,23] as $P = 1.98 \times 10^{-24}$ (binomial test). This observation corresponds to an odds ratio (OR) of 11.1 (95% confidence interval (CI) $= 7.7$–15.4) compared with null expectations strongly supporting the enrichment of ASD candidates in our 189 gene set. We repeated this analysis removing the 19 known ASD/ID genes (Supplementary Data 1) and observed a reduced but still highly significant signal ($P = 1.17 \times 10^{-5}$) corresponding to an odds ratio of 4.1 (95% CI $= 2.2$–7.0). The remaining validated variants ($n = 278$) were either inherited or inheritance status could not be determined. We observed a slight (1.1) bias for maternally inherited LGD variants (116 maternal versus 104 paternal) among the 1,045 trios. Among the DN mutations, there were 30 mutations (22 LGD and 8 MIS30) in 15 genes with published recurrent LGD mutations identified from the SSC and ASC cohorts (ADNP, ASH1L, CHD2, CHD8, DSCAM, DYRK1A, GRIN2B, NCKAP1, MED13L, POGZ, RIMS1, SCN2A, SYNGAP1, TRIP12, WDFY3; Table 1). Within the ACGC, we observed recurrent DN mutations in seven genes: SCN2A, CHD8, DSCAM, MECP2, POGZ, WDFY3 and ASH1L (Table 1). The most frequently mutated gene was SCN2A with eight total severe DN mutations (seven LGD and one MIS30) representing 16.2% (43/265) of all severe DN proband mutations in this study.

To further refine the total DN mutation rate for our top risk genes, we evaluated all rare missense mutations in the 29 genes where a DN LGD or DN MIS30 event had been identified in the

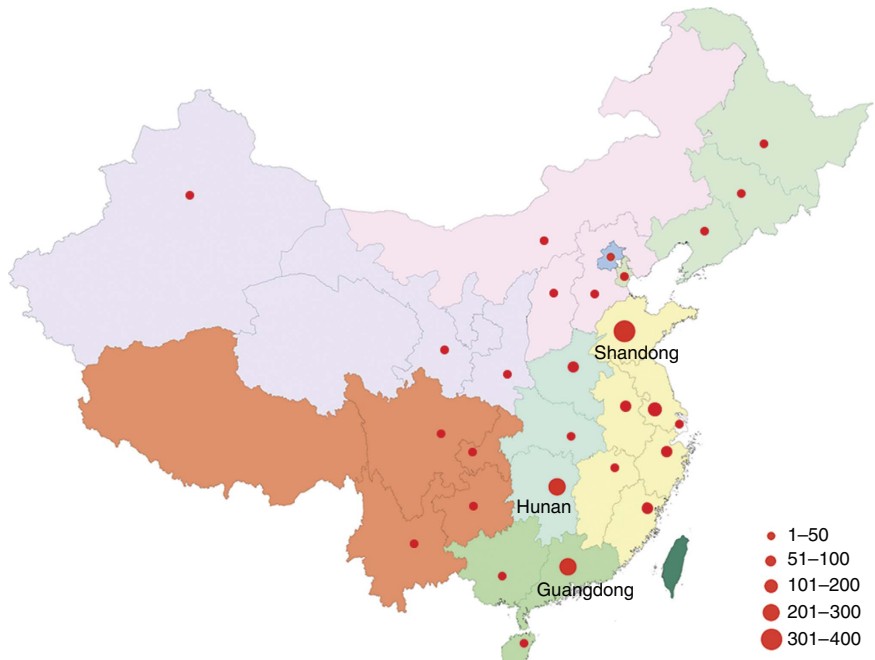

**Figure 1 | Birthplace distribution of ASD cases in the ACGC.** The patients involved in this study are distributed throughout China with the majority recruited from Shandong ($n = 389$), Hunan ($n = 264$) and Guangdong ($n = 202$) provinces. The map was generated from a template downloaded from the public standard map service (http://219.238.166.215/mcp/index.asp) of the National Administration of Surveying, Mapping and Geoinformation of China (http://www.sbsm.gov.cn). The template is freely available for public download and use. Different colours represent different geographical regions, provinces or municipalities in China. The size of the red circles denotes the relative number of patients from each locale. Editor's Note: *Nature Communications* remains neutral with regard to jurisdictional claims in published maps.

ACGC (Supplementary Data 4). We validated all missense events with a CADD score $\leq 30$ (522 events; Supplementary Data 5; Supplementary Table 3; Supplementary Fig. 3 shows the distribution of CADD scores among the 29 genes). From the events that validated in the proband by Sanger sequencing (429/522; 82.2%), we identified 11 (2.8%) new DN missense mutations from samples where parental DNA were available ($n = 386$). This is in contrast to the MIS30 events where 21.6% (8/37) of mutations were DN. While CADD cutoffs are certainly an imperfect definition of pathogenicity, such thresholds significantly increase the odds that a DN and likely disease-causing mutation will be found (CADD $> 30$, OR $= 8.77$, $P = 7.83 \times 10^{-5}$, Fisher's exact test). Among the low CADD DN missense events, 5 of the 11 were associated with our top two genes (*SCN2A* and *CHD8*). We identified four additional missense DN mutations (CADD $\leq 30$) in *SCN2A*, which increased the total number to 12 DN mutations (seven LGD and five missense; Fig. 2). We also identified one additional DN missense mutation in *CHD8* (for a total of three LGD mutations and one missense mutation; Fig. 2). DN mutations in *SCN2A*, thus, account for approximately 1.1% of probands in our Chinese cohort.

**Comparison of DN LGD events between populations**. To determine whether the top DN ASD risk genes differ between European and Chinese populations, we extracted the regions for the 189 genes targeted in this study from exome sequencing data from 2,508 SSC families (1,911 quads and 597 trios) and 1,445 ASC trio families (Supplementary Data 6). We compared the total DN LGD mutation rates for these 189 genes between Chinese and European probands (1,892 SSC probands where both parents were self-reported in the Caucasian ethnic group); no significant difference was observed ($P = 0.15$, Fisher's exact test, OR $= 0.74$, 95% CI $= 0.48$–1.11). When all the SSC and ASC samples were

considered regardless of self-reported ethnicity, there was still no significant difference observed compared with the Chinese cohort ($P = 0.22$, Fisher's exact test, OR $= 0.79$, 95% CI $= 0.53$–1.15) supporting our null hypothesis that the overall rate of DN mutations would be constant between different ethnic groups (Supplementary Discussion).

Notably, the rate of *SCN2A* DN LGD mutations appears to be increased in our study (7/1,045; 0.7%) compared with published exome sequencing studies of ASD families, specifically the SSC and ASC (4/3,953; 0.1%). This difference was nominally significant ($P = 2.6 \times 10^{-3}$, two-tailed Fisher's exact test, OR $= 6.7$, 95% CI $= 1.7$–31.1) but did not withstand multiple testing correction for all 189 candidate genes. Compared with published MIP-based studies of individuals with ID/DD (3/3,387 or 0.09% of individuals carried an *SCN2A* LGD mutation)[20], our data still appear to trend toward increased numbers of individuals carrying DN LGD *SCN2A* mutations ($P = 2.4 \times 10^{-3}$). Because DN and private LGD mutations in candidate risk genes are individually rare (Supplementary Fig. 4 and Supplementary Data 9), our current study is underpowered to robustly detect differences in mutation frequencies between affected populations at the single-gene level. However, this nominally significant finding is of interest for potential validation in future studies of larger sample cohorts (Supplementary Discussion).

**DN mutation recurrence among Chinese ASD probands**. As part of this study design, we selected candidate genes with as few as one DN LGD mutation in samples of primarily European descent for resequencing in Chinese samples with the aim of identifying additional mutations, thus establishing DN LGD recurrence for these candidate ASD risk genes. We identified DN LGD mutations in *CUL3*, *DOCK8*, *GIGYF2*, *MYT1L* and *ZNF292* (Table 2), establishing recurrence for these loci where only one LGD event had previously been identified in the SSC and ASC.

**Table 1 | Comparison of DN mutations in the ACGC versus SSC and ASC.**

| Gene ID | ACGC (n = 1,045) | | | | | SSC (n = 2,508) | | ASC (n = 1,445) | | All (n = 4,998) | | |
|---|---|---|---|---|---|---|---|---|---|---|---|---|
| | LGD | MIS (>30/≤30) | ACGC DN (LGD/MIS) | ACGC DN P values | ACGC DN q values | LGD | MIS (>30/≤30) | LGD | MIS (>30/≤30) | ALL DN (LGD/MIS) | ALL DN P values | ALL DN q values |
| SCN2A | 7 | 5 (1/4) | 12 (7/5) | 2.53E − 22 | 4.79E − 20 | 2 | 4 (0/4) | 2 | 4 (3/1) | 24 (11/13) | 4.02E − 34 | 7.59E − 32 |
| CHD8 | 3 | 1 (0/1) | 4 (3/1) | 4.52E − 06 | 0.00042723 | 9 | 0 (0/0) | 0 | 2 (1/1) | 15 (12/3) | 1.39E − 17 | 1.32E − 15 |
| DSCAM | 2 | 1 (0/1) | 3 (2/1) | 0.00186172 | 0.03746743 | 3 | 1 (0/1) | 1 | 0 (0/0) | 8 (6/2) | 2.49E − 05 | 0.000314 |
| MECP2 | 2 | 0 (0/0) | 2 (2/0) | 0.00064298 | 0.0243046 | 0 | 0 (0/0) | 1 | 1 (0/1) | 4 (3/1) | 3.29E − 05 | 0.00036 |
| WDFY3 | 1 | 2 (1/1) | 3 (1/2) | 0.00102697 | 0.03234943 | 2 | 2 (0/2) | 0 | 2 (0/2) | 9 (3/6) | 5.7E − 07 | 1.54E − 05 |
| ADNP | 1 | 1 (0/1) | 2 (1/1) | 0.00196919 | 0.03746743 | 2 | 0 (0/0) | 2 | 0 (0/0) | 6 (5/1) | 8.88E − 07 | 2.1E − 05 |
| DYRK1A | 1 | 1 (0/1) | 2 (1/1) | 0.00185109 | 0.03746743 | 4 | 0 (0/0) | 1 | 0 (0/0) | 7(6/1) | 3.12E − 08 | 1.18E − 06 |
| GIGYF2 | 1 | 1 (0/1) | 2 (1/1) | 0.0019824 | 0.03746743 | 1 | 1 (0/1) | 0 | 1 (1/0) | 5 (2/3) | 1.78E − 05 | 0.000271 |
| POGZ | 1 | 1 (1/0) | 2 (1/1) | 0.00019851 | 0.01250592 | 2 | 3 (0/3) | 1 | 0 (0/0) | 8 (4/4) | 1.63E − 13 | 1.03E − 11 |
| ARHGAP32 | 1 | 0 (0/0) | 1 (1/0) | 0.16546346 | 1 | 0 | 0 (0/0) | 0 | 1 (0/1) | 2 (1/1) | 0.214733 | 0.369991 |
| CDKL5 | 1 | 0 (0/0) | 1 (1/0) | 0.1325453 | 1 | 0 | 0 (0/0) | 0 | 0 (0/0) | 1 (1/0) | 0.493392 | 0.630075 |
| CUL3 | 1 | 0 (0/0) | 1 (1/0) | 0.0395099 | 0.67885192 | 1 | 0 (0/0) | 0 | 0 (0/0) | 2 (2/0) | 0.016357 | 0.055725 |
| DOCK8 | 1 | 0 (0/0) | 1 (1/0) | 0.33411074 | 1 | 0 | 0 (0/0) | 1 | 0 (0/0) | 2 (2/0) | 0.578781 | 0.719669 |
| GRIN2B | 1 | 0 (0/0) | 1 (1/0) | 0.10955634 | 1 | 3 | 1 (0/1) | 0 | 0 (0/0) | 5 (4/1) | 0.000277 | 0.001967 |
| MED13L | 1 | 0 (0/0) | 1 (1/0) | 0.11967793 | 1 | 2 | 1 (0/1) | 0 | 0 (0/0) | 4 (3/1) | 0.003549 | 0.0156 |
| MYT1L | 1 | 0 (0/0) | 1 (1/0) | 0.13071868 | 1 | 0 | 1 (0/1) | 1 | 1 (0/1) | 4 (2/2) | 0.004939 | 0.020294 |
| NCKAP1 | 1 | 0 (0/0) | 1 (1/0) | 0.0564522 | 0.82072814 | 2 | 0 (0/0) | 0 | 0 (0/0) | 3 (3/0) | 0.002907 | 0.013402 |
| NCOR1 | 1 | 0 (0/0) | 1 (1/0) | 0.11395926 | 1 | 0 | 1 (0/1) | 0 | 0 (0/0) | 2 (1/1) | 0.114906 | 0.246787 |
| PHIP | 1 | 0 (0/0) | 1 (1/0) | 0.16104015 | 1 | 0 | 1 (0/1) | 0 | 0 (0/0) | 2 (1/1) | 0.205545 | 0.366491 |
| RIMS1 | 1 | 0 (0/0) | 1 (1/0) | 0.10950747 | 1 | 2 | 0 (0/0) | 0 | 0 (0/0) | 3 (3/0) | 0.018868 | 0.062563 |
| SHANK1 | 1 | 0 (0/0) | 1 (1/0) | 0.44442526 | 1 | 0 | 1 (0/1) | 0 | 0 (0/0) | 2 (1/1) | 0.770738 | 0.933778 |
| STXBP1 | 1 | 0 (0/0) | 1 (1/0) | 0.04797519 | 0.7556092 | 0 | 0 (0/0) | 0 | 0 (0/0) | 1 (1/0) | 0.209526 | 0.366671 |
| SYNGAP1 | 1 | 0 (0/0) | 1 (1/0) | 0.27578289 | 1 | 1 | 2 (0/2) | 4 | 1 (0/1) | 9 (6/3) | 3.43E − 05 | 0.00036 |
| TRIP12 | 1 | 0 (0/0) | 1 (1/0) | 0.08225617 | 1 | 1 | 2 (2/0) | 0 | 0 (0/0) | 4 (2/2) | 0.000853 | 0.004888 |
| ZNF292 | 1 | 0 (0/0) | 1 (1/0) | 0.18344779 | 1 | 1 | 2 (0/2) | 0 | 0 (0/0) | 4 (2/2) | 0.074708 | 0.185788 |
| ASH1L | 0 | 3 (2/1) | 3 (0/3) | 0.00039966 | 0.01888397 | 2 | 0 (0/0) | 0 | 1 (0/0) | 6 (3/3) | 6.69E − 05 | 0.000602 |
| CHD2 | 0 | 1 (1/0) | 1 (0/1) | 0.08927943 | 1 | 4 | 0 (0/0) | 0 | 2 (1/1) | 7 (4/3) | 4.8E − 07 | 1.51E − 05 |
| ITPR1 | 0 | 1 (1/0) | 1 (0/1) | 0.36341406 | 1 | 0 | 1 (0/1) | 0 | 1 (0/1) | 3 (0/3) | 0.366449 | 0.501876 |
| TSC2 | 0 | 1 (1/0) | 1 (0/1) | 0.32552866 | 1 | 0 | 3 (0/3) | 0 | 0 (0/0) | 4 (0/4) | 0.122424 | 0.252273 |

LGD, de novo likely gene-disruptive mutation; MIS > 30, de novo missense mutations with CADD score greater than 30; MIS ≤ 30, de novo missense mutations with CADD score less than or equal to 30. Nominal P values and q values were from DN mutation simulation, the probabilistic framework developed previously[14,23]. The combined data set represents data from 4,998 autism patients.

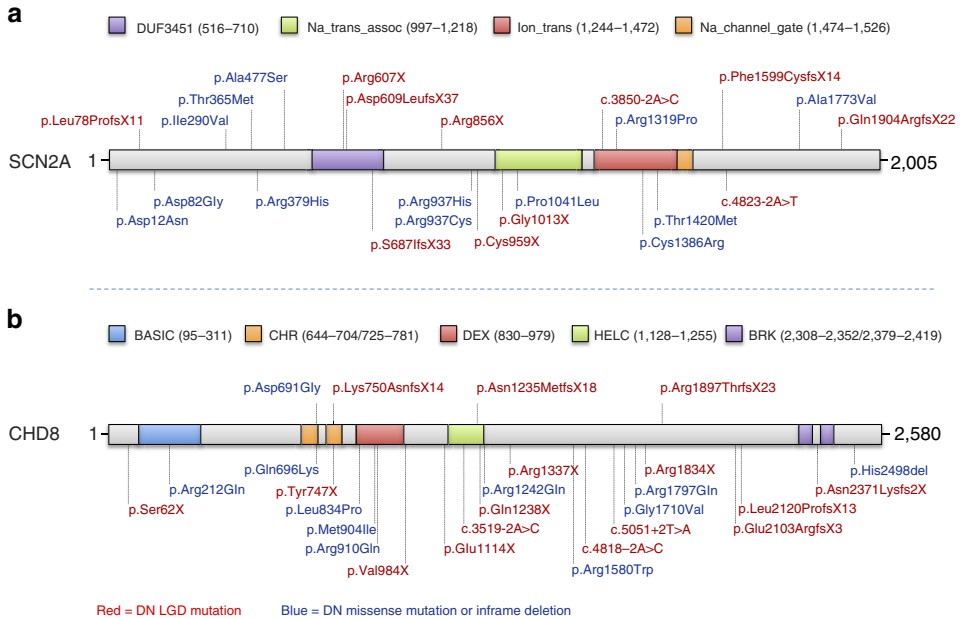

**Figure 2 | Protein diagram of SCN2A and CHD8 including gene mutation locations.** (**a**) SCN2A mutations above the diagram were identified in ACGC samples, including seven DN LGD mutations and five DN missense mutations (DN frequency = 1.1%). Mutations below the protein diagram were identified in the SSC and ASC samples, including four DN LGD mutations and eight DN missense mutations (DN frequency = 0.3%). (**b**) CHD8 mutations above the diagram were identified in ACGC samples, including three DN LGD mutations and one DN missense mutation. Mutations below the protein diagram were identified in previous large cohort ASD studies, including 13 DN LGD mutations and 10 DN missense mutations.

**Table 2 | DN mutations in ACGC and exome cohorts (SSC and ASC) of novel implicated risk genes in this study.**

| Gene ID | ACGC | | | | SSC and ASC | | ALL p_LGD | ALL q_LGD | DN_CNVs | Combined (LGD/MIS/CNV) |
|---|---|---|---|---|---|---|---|---|---|---|
| | LGD | MIS (>30/≤30) | p_LGD | q_LGD | LGD | MIS (>30/≤30) | | | | |
| *From single LGD to recurrent LGD* | | | | | | | | | | |
| CUL3 | 1 | 0 (0/0) | 0.00445 | 0.13202 | 1 | 0 (0/0) | 0.000224515 | 0.001672541 | 0 | 2 (2/0/0) |
| DOCK8 | 1 | 0 (0/0) | 0.03997 | 0.31474 | 1 | 0 (0/0) | 0.016720467 | 0.059625815 | 2 | 4 (2/0/2) |
| GIGYF2 | 1 | 1 (0/1) | 0.01015 | 0.14894 | 1 | 2 (1/1) | 0.001152942 | 0.006402713 | 0 | 4 (2/2/0) |
| MYT1L | 1 | 0 (0/0) | 0.01252 | 0.15706 | 1 | 2 (0/2) | 0.00174452 | 0.009158728 | 0 | 4 (2/2/0) |
| TRIP12 | 1 | 0 (0/0) | 0.01024 | 0.14894 | 1 | 2 (2/0) | 0.001173356 | 0.006402713 | 1 | 5 (2/2/1) |
| ZNF292 | 1 | 0 (0/0) | 0.01992 | 0.18823 | 1 | 2 (0/2) | 0.09173358 | 0.176914761 | 0 | 4 (2/2/0) |
| | | | | | | | | | | |
| *From CNVs to individual gene(s)* | | | | | | | | | | |
| ARHGAP32 | 1 | 0 (0/0) | 0.01703 | 0.16938 | 0 | 1 (0/1) | 0.078852143 | 0.161989728 | 1 | 3 (1/1/1) |
| NCOR1 | 1 | 0 (0/0) | 0.0148 | 0.15706 | 0 | 1 (0/1) | 0.068806647 | 0.144428043 | 1 | 3 (1/1/1) |
| | | | | | | | | | | |
| *Novel genes with DN LGD mutation* | | | | | | | | | | |
| SHANK1 | 1 | 0 (0/0) | 0.04871 | 0.36822 | 0 | 1 (0/1) | 0.212424752 | 0.368332827 | 0 | 2 (1/1/0) |
| STXBP1 | 1 | 0 (0/0) | 0.00489 | 0.13202 | 0 | 0 (0/0) | 0.023169362 | 0.0742205 | 0 | 1 (1/0/0) |
| PHIP | 1 | 0 (0/0) | 0.02421 | 0.20799 | 0 | 1 (0/1) | 0.110602121 | 0.204939224 | 0 | 2 (1/1/0) |
| CDKL5 | 1 | 0 (0/0) | 0.01496 | 0.15706 | 0 | 0 (0/0) | 0.069539428 | 0.144428043 | 0 | 1 (1/0/0) |

We define novel DN LGD recurrence as a second DN LGD mutation in the ACGC in addition to the SSC. LGD, MIS > 30 and MIS ≤ 30 have the same meaning as in Table 1. p_LGD and q_LGD are P and q values from DN LGD mutation simulation using a probabilistic framework developed previously[14,23]. ALL p_LGD and ALL q_LGD are the P and q values for all DN LGD mutation simulations combined from the ACGC, SSC and ASC data sets.

The identification of two DN LGD mutations in the combined sample size of the ACGC and SSC is unlikely[9]. Of these five genes, *MYT1L* shows no evidence of rare LGD mutations in a large population control (ExAC; $n = 45,376$), while two genes had a small number of rare LGD mutations in this control population (*GIGYF2* had nine individuals carrying rare LGD mutations at three sites and *CUL3* had four individuals carrying private LGD variants). None of these three genes showed evidence of common LGD variation in the ExAC cohort (Supplementary Data 7 and 11). In addition, we also integrated our DN results from the ACGC with DN CNVs identified from SSC and Autism Genome Project (AGP) families. From these candidate CNVs, we identified a DN LGD mutation in both *ARHGAP32* and *NCOR1* (Table 2, Fig. 3). Finally, we identified one DN LGD mutation in each of four genes—*SHANK1*, *PHIP*, *STXBP1* and *CDKL5*—of which, *SHANK1*, *STXBP1* and *CDKL5* are already known to be associated with syndromic and non-syndromic forms of ASD (Table 2).

To further investigate the potential impact of rare inherited LGD variants, we compared our rare LGD event rates with those from the ExAC cohort (using the subset of 45,376 neuropsychiatric disease-free individuals). We combined these data with CNV data from 29,085 children with ID/DD and 19,584 controls using a joint probability model previously described in Coe et al.[20]. Briefly, rates of LGD mutations and deletion CNVs between proband and control cohorts are combined using a statistical model that assumes array and MIP patients are independently assayed. Under this model, we examined 57 autosomal genes with at least one LGD mutation and sufficient sequence coverage in both exome and MIP samples. We identified a significant enrichment ($q < 0.05$) for LGD events for 10 targets (Supplementary Data 7), including three genes with only a single LGD event identified in our study. While these 10 genes are good candidates, most CNVs are large and encompass many genes; in addition, the presence of overlapping large CNVs and DN LGD events does not confer the same specificity as recurrent LGD mutations.

Because there are 498 probands with only one or no parental sample(s) available, inheritance for variants in this subset cannot

be determined. Several LGD mutations in these samples have a high probability of being DN considering the low-predicted mutability and evidence of DN mutations in these genes in European ASD cohorts. In addition, our integrated CNV and LGD burden analysis highlight genes with an increased likelihood of haploinsufficiency in ID/DD/ASD that do not reach significance in this cohort by analysis of DN single-nucleotide variants and indels alone. These variants include a frameshift mutation in *GIGYF2* in proband M15067, a stop-gain mutation in *STXBP1* in proband M17663, and a frameshift mutation in *SYNGAP1* in proband M19759. We also identified three LGD mutations in *RIMS1* with undetermined inheritance. Counts of all LGD and missense mutations by gene are provided (Supplementary Data 8).

**Clinical characterization of genotypes and phenotypes**. We attempted to recontact all patients with DN mutations identified in this study to collect detailed phenotypic information. In the end, 20 patients (~46.5% of patients with DN mutations) were successfully recontacted and detailed phenotypes were recorded (Table 3). We found that ID (15/16), gastrointestinal (GI) disturbance (9/20), and macrocephaly (5/15) are some of the most common comorbid conditions among these patients.

Although the number of individual patients for any gene is few, several of the phenotypes identified during patient recontact were reminiscent of those reported in the literature. The patients with the *CHD8* DN mutation, for example, show ID (2/2), macrocephaly (2/2), tall stature (2/2), high BMI (2/2), mild regression (2/2), sleep problems (1/2), GI disturbance (1/2), attention deficit (2/2) and anxiety (1/2). These findings are consistent with the clinical and genetic subtype previously reported for this gene[24]. Similarly, patients with the *SCN2A* DN mutation show other comorbid conditions, including ID (3/3) and seizures (2/3) consistent with the observation that *SCN2A* DN mutations have been identified among ID and epileptic encephalopathy patient cohorts[21,22,25]. Among our patients with *ADNP* and *DYRK1A* DN mutations, we noted congenital cardiac defects. In addition, the *ADNP* proband also has attention problems, and the *DYRK1A*

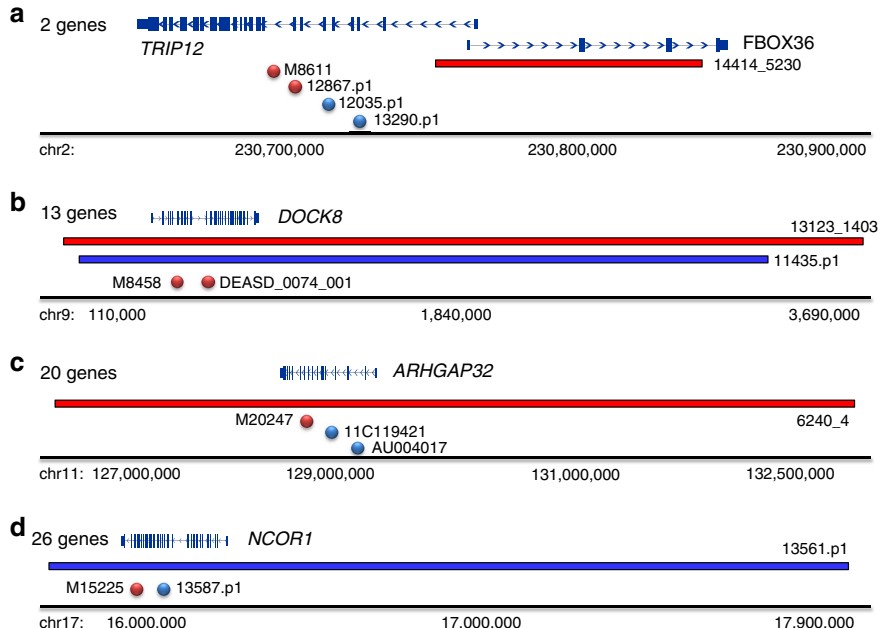

**Figure 3 | CNV and single-nucleotide variant DN mutations identify likely autism risk genes.** DN mutation patterns from SSC, AGP and ACGC samples for genes (**a**) TRIP12, (**b**) DOCK8, (**c**) ARHGAP32 and (**d**) NCOR1 identify the most likely candidate genes from larger pathogenic CNV interval. Gene numbers denote the number of RefGene entries in each corresponding CNV interval. CNV deletions (red horizontal bars) and duplications (blue bars) are shown with respect to DN LGD (red circles) and DN missense (blue circles) mutations.

proband reported febrile seizures in infancy, hypertonia, neonatal feeding problems and sleep disturbances. All of these phenotypes have been described before in association with DN mutation of these genes[26,27].

Several other novel comorbidities associated with CHD8, ADNP, POGZ, ARHGAP32, GRIN2B, ASH1L, NCKAP1, CHD2 and DSCAM are also recorded (Table 3). Larger cohorts of patients with DN mutations in these genes will be required to assess the significance of these observations.

## Discussion

We have performed an investigation of DN mutations among ASD candidate risk genes in a Chinese ASD patient cohort. Among the 1,045 trios tested, we discovered 43 DN mutations in 29 of the 189 candidate genes queried by our MIP-based approach. The most frequently DN mutated gene in our study was SCN2A where we identified seven novel DN LGD mutations and five DN missense mutations. DN mutations in SCN2A accounted for ~1.1% of ASD patients in our Chinese cohort. This rate was higher than exome- and MIP-sequenced ASD cohorts of European descent. This observation is not likely explained by differences in DN mutation rates among global populations or paternal age but rather by ascertainment and technology biases. For example, published head-to-head comparisons of exome and MIP sequencing technologies on the SSC cohort have shown that MIPs have greater sensitivity for variant detection[14], which could also account for these increased rates of DN variation in our MIP study compared with exome sequencing studies.

Many sequencing studies of ASD and ID/DD patients have highlighted the extensive overlap among mutated genes between these comorbid conditions[4,21,22,28]. Indeed, it has been estimated that >40% of patients with ASD may suffer some form of cognitive impairment[29]. Even among some of the most rigorously ascertained autism cohorts, such as the SSC, it estimated that 30% of SSC patients also have ID[30]. Although all of the ACGC patients

met strict autism (DSM-IV) criteria, IQ data was not routinely collected for the majority of patients in this study. It is possible that some of the most severely affected individuals may have been initially recruited from the referring centres. Our patient follow-up for specific mutations generally supports this hypothesis. For example, four of the SCN2A families were available for recontact and three of these families had available IQ data that showed impaired cognition for the proband. Further, one of these families also reported severe language impairment adding credence to our hypothesis that this cohort, although defined as autistic by the DSM-IV, is enriched for patients who are also cognitively impaired[31]. This is consistent with studies of autism families in the SSC where three of the original four patients identified with DN SCN2A mutations were also intellectually impaired (IQ<70)[12].

The ability to recontact patients with DN mutations in this study was important both for confirming previously published genotype–phenotype correlations as well as describing potentially new genetic subtypes of ASD. For SCN2A, in addition to ID in three of these patients, two of the four families that could be recontacted also reported seizures; SCN2A has been previously implicated in epilepsy[32]. We also identified three patients with DN CHD8 LGD mutations. Two of them were recontacted and the phenotypes were similar to those previously reported[24]. Similarly, mutations in ADNP are reportedly linked to a shared clinical phenotype, including ID, facial dysmorphism and congenital heart defects[27]. The ADNP proband in this study had severe ID, a congenital heart defect, attention problems, mild regression and differently sized eyes. The patients with the DN DYRK1A LGD mutation also presented similar phenotypes as described previously[26], such as ID, febrile seizures in infancy, hypertonia, neonatal feeding problems, cardiac defects at birth and sleep disturbances. Microdeletions of SHANK1 have been previously identified in ASD patients[33]. In addition to an ASD diagnosis, our SHANK1 patient also presented with ID, typical regression (after 3–4 years old) and an abnormal gait.

We also report here clinical phenotypes for anecdotal cases of DN LGD mutations in WDFY3, GIGYF2, MED13L, NCKAP1,

**Table 3 | Genotype and phenotype correlations of DN mutations.**

| Patient ID | M8856 | M15222 | M21522 | M23243 | M8656 | M17463 | M18477 | M19813 | M20247 | M20274 |
|---|---|---|---|---|---|---|---|---|---|---|
| Gene Mutation | SCN2A c.3850-2A>C | SCN2A p.Arg1319Pro | SCN2A p.Gln1904Argfs*22 | SCN2A p.Phe1599Cysfs*14 | CHD8 p.Lys750Asnfs*14 | CHD8 p.Arg1897Thrfs*23 | WDFY3 p.Gln1760* | POGZ p.Gln127* | ARHGAP32 p.Glu1223Lysfs*22 | DYRK1A p.Glu153* |
| Sex | M | M | M | M | M | F | M | F | F | M |
| ASD | + | + | + | + | + | + | + | + | + | + |
| Intellectual disability | Severe | Severe | Moderate | NA | Severe | 33 | 50–60 | Moderate | 61 | NA |
| Developmental delay (motor) | + | + | NA | − | + | + | + | + (mild) | + | NA |
| Developmental delay (speech) | + | + | + | + | + | + | + | + | + | + |
| Repetitive behaviour | + | + | + | + (mild) | + | + | + | + | + | + |
| Macrocephaly | − | − | NA | − | + | + | + | − | NA | − |
| Microcephaly | − | − | NA | − | − | − | − | − | NA | + |
| EEG | − | + | − | − | − | − | + | + | NA | − |
| MRI | − | − | + | + | − | − | + | − | NA | + |
| Regression | + (mild) | + (mild) | − | NA | + (mild) | + (mild) | − | − | − | − |
| Seizures | + | + | − | − | − | − | − | + | − | − |
| Sleep problems | + | − | + | + | − | + | + | − | − | + |
| GI disturbances | − | − | − | + | + | − | + | − | + | + |
| Hypotonia | − | − | NA | − | + | − | − | − | − | NA |
| Hypertonia | + | − | NA | − | − | − | − | − | − | NA |
| Hyperactive behaviour | − | + | + | − | − | − | + | − | − | + |
| Attentional problems | + | + | + | − | + | + | + | + | + | − |
| Anxiety | + | − | + (mild) | − | + | − | + | − | − | NA |
| Aggressive behaviour | − | + | − | − | − | − | − | − | − | − |
| Obsessive behaviour | − | − | − | − | − | − | + | + | − | − |
| Febrile seizures infancy | − | − | + | − | − | − | − | − | − | + |
| C-section | − | − | NA | + | − | + | + | − | − | + |
| Premature birth | − | − | − | − | − | − | − | − | − | + |
| Neonatal feeding problems | − | − | NA | + | − | − | − | − | − | + |
| Childhood feeding problems | − | − | NA | + | − | − | NA | − | − | + |
| Karyotype | NA | 46, XY | NA | 46, XY | na | 46, XX | NA | 46, XX | 46, XX | 46, XY |
| Specific phenotypes | − | − | − | Special talents at computer | Tall, overweight | Tall, overweight | − | − | − | Congenital heart defect, IUGR |

| PatientID | M20277 | M20699 | M20837 | M23139 | M23196 | M23209 | M23688 | M23762 | M27851 | M27882 |
|---|---|---|---|---|---|---|---|---|---|---|
| Gene Mutation | GRIN2B p.Arg519* | MYT1L p.Gln848* | ASH1L p.Val2080Ile | SHANK1 c.2458+1G>A | MED13L p.Gln799Glyfs*10 | NCKAP1 c.3180+1G>A | CHD2 p.Arg1685His | GIGYF2 p.Glu320* | DSCAM p.Pro356Leufs*5 | ADNP p.Leu211* |
| Sex | M | M | M | F | F | M | M | M | M | M |
| ASD | + | + | + | + | + | + | + | + | + | + |
| Intellectual disability | Severe | 60 | <30 | Moderate | NA | − | 38 | Mild | NA | 45 |
| Developmental delay (motor) | + | + | na | + | + | + | NA | + | NA | + |
| Developmental delay (speech) | + (mild) | + | + | + | + | + | + | + | + | + |
| Repetitive behaviour | + | + | + | + | + (mild) | + | + | + | + | + |
| Macrocephaly | − | − | + | − | − | − | NA | + | NA | NA |
| Microcephaly | − | − | − | − | − | − | NA | − | NA | NA |
| EEG | − | − | − | + | − | − | − | NA | NA | − |
| MRI | − | + | − | − | + | − | + | NA | − | − |
| Regression | + (mild) | − | + (mild) | + (typical) | − | − | − | − | − | + (mild) |
| Seizures | − | − | − | − | − | − | − | − | − | − |
| Sleep problems | − | + | + | + | + | − | + | − | − | − |
| GI disturbances | + | − | + | − | − | + | + | − | − | − |
| Hypotonia | − | − | − | − | + | − | NA | NA | NA | NA |
| Hypertonia | + | + | − | − | − | − | NA | NA | NA | NA |
| Hyperactive behaviour | + | − | + | − | − | + | + | − | + | − |
| Attentional problems | − | + | + | + | − | − | +? | + | + | + |
| Anxiety | + | + | + | + | − | − | + | + | − | − |
| Aggressive behaviour | − | − | − | − | − | − | + | − | − | − |
| Obsessive behaviour | + | − | + | − | − | − | + | − | + | − |
| Febrile seizures infancy | − | + | − | − | + | − | − | − | − | − |
| C-section | − | − | − | + | + | + | − | − | + | − |
| Premature birth | − | − | − | + | − | − | − | − | − | − |
| Neonatal feeding problems | + | − | − | − | − | − | − | − | − | − |
| Childhood feeding problems | − | − | − | − | − | − | − | − | − | NA |
| Karyotype | 46, XY | 46, XY | 46, XY | 46, XX | 46, XX | 46, XY | 46, XY, t(8;16)(q24.3;q22)pat | 46, XY | 46, XY | 46, XY |
| Specific phenotypes | − | Walk pigeon Tic syndrome | − | Typical regression Walk pigeon | Mandibular protrusion hyperopia | Good prognosis | − | − | − | Congenital heart defect, different size of both eyes |

NA, not available.

*ARHGAP32*, *DSCAM* and *MYT1L*, which serves as a useful starting point for further phenotype–genotype correlations. In addition, we also noted specific comorbidities in several patients that have not been previously described. The patient with a *MYT1L* DN mutation was diagnosed with a motor tic disorder and showed an unusual gait. The patient with a *MED13L* DN mutation has hyperopia and a clear dysmorphia typified by mandibular protrusion. The patient with a *WDFY3* DN LGD mutation has ID and macrocephaly and had severe GI disturbances up to 4 years of age, as well as hyperactivity, sleep problems, attention problems and anxiety. The patient with a *GIGYF2* DN LGD mutation also has mild ID and macrocephaly. His mother reported an obsession with food but that he has no sense of his appetite being satiated in addition to problems with attention and anxiety.

We note that *MED13L* haploinsufficiency is well established[34,35]; others, such as *DSCAM* and *GIGYF2*, are emerging high-impact risk genes from exome and CNV studies[23,36,37]. For example, Akshoomoff and colleagues recently highlighted the neuron-associated GTPase activating protein (*ARHGAP32*) as the best candidate gene for ASD features associated with Jacobsen syndrome based on overlapping deletions that narrowed the critical region to include this locus among three other genes[38]. To our knowledge, we discovered the first DN LGD mutation in *ARHGAP32* in a patient with ASD although one DN missense variant has been previously reported[13]. Similarly, the discovery of a patient with a DN LGD mutation in *NCOR1* (Table 3), now in addition to published DN missense mutations in ASD[12] and DD[28] cohorts, is exciting in light of the known interaction of the Rett syndrome gene (*MECP2*) with members of this nuclear receptor co-repressor complex[39,40].

This study confirms the importance of DN mutations in ASD and further shows that candidate risk genes originally identified primarily from European ASD cohorts are highly relevant in other populations, such as the Chinese cohort described here. The large number of DN events identified in *SCN2A* (both LGD and missense) confirm the significance of DN missense variation. There will likely be risk genes where both LGD and missense mutations contribute to disease etiology, perhaps in the case of *SCN2A*. Other risk genes may predominantly be represented by DN LGD events (for example, *CHD8* and *ADNP*) or missense events. Larger numbers of samples will need to be screened to identify the diversity of risk genes that contribute to disease variation. Detailed clinical follow-up will continue to be essential in this genotype-first model, particularly for risk genes where both DN LGD and missense signals significantly associate with disease phenotypes, to determine whether different classes of mutations have different biological effects (for example, haploinsufficient versus dominant-negative effects)[41]. These efforts will be very beneficial for the early disease warning and diagnosis and, most importantly, early intervention while also providing the motivation for further functional and translational studies of disease risk genes, which will aid in future personalized treatments. Unlike rare and common variants that are frequently skewed in their population distribution, our study raises the exciting possibility that extremely large autism cohorts may in the near future be amassed across the continents to prove the pathogenicity of autism disease genes in cohorts of hundreds of thousands of patients.

## Methods

**Human subjects.** All the subjects who participated in this study completed informed consent before the original sample collection. The DNA samples were extracted from the peripheral blood of patients and parents if available. Probands were diagnosed with ASD based on DSM-IV criteria, and families were excluded if the proband did not meet these diagnostic criteria. Detailed DSM-IV diagnostic criteria are described in Supplementary Table 1. Autism-related single-gene disorders, such as fragile X syndrome, tuberous sclerosis complex and phenylketonuria, were excluded where possible. While patients originate from across China, the majority of samples came from three provinces: namely, Shandong, Hunan and Guangdong (see Supplementary Table 2 for sample distribution). The recontacted patients underwent a thorough assessment, including a detailed review of medical history and comprehensive physical and neurocognitive phenotype. Instruments such as the ABC (autism behaviour checklist) and SRS (social responsiveness scale) were applied where possible. This study was approved by the Institutional Review Board (IRB) of the State Key Laboratory of Medical Genetics, School of Life Sciences at Central South University, Changsha, Hunan, China and adhered to the tenets of the Declaration of Helsinki. In addition to approval by the local IRBs, the study has been reviewed and is compliant with the Chinese Ministry of Science and Technology (MOST) for the Review and Approval of Human Genetic Resources (Approval number: HGR-2016-10-55:556).

**MIPs and pooling.** MIPs were designed using MIPgen with an updated scoring algorithm[42]. MIPs were composed of a common linker, which is 30 base pairs (bp), flanked by extension and ligation arms ranging from 15 to 30 bp (the total MIP length is between 75 and 80 bp). Each MIP with unique arms will target a specific genomic region and set to a total fixed length of 162 bp. Oligonucleotides were ordered from Integrated DNA Technologies (IDT, Coralville, IA, USA; MIPs listed in Supplementary Data 2). Five degenerate bases were added between the common linker and the extension arm, which allow a non-duplicate coverage of $4^5 = 1,024$ as the theoretical maximum. In cases of polymorphisms that may interfere with capture, two MIPs were designed to capture either haplotype. MIPs were designed against the GRCh37 human genome reference using dbSNP138. MIPs were pooled together by gene. For initial testing (1X pool), probes were combined at equal molar concentrations and phosphorylated. After initial testing, probes that performed poorly were repooled and phosphorylated with increased molar ratios at 10X or 50X to rescue the capture coverage. The final pools (1X, 10X and 50X) were combined as a working pool. For the purpose of sequencing, we divided the smMIPs into three pools; we tested and rebalanced each pool independently using 16 unaffected (HapMap) samples as controls. We distinguished the three pools by name as SKLMG-PL1 (63 genes with 3,557 MIPs total), SKLMG-PL2 (71 genes with 3,773 MIPs total) and SKLMG-PL3 (55 genes with 3,563 MIPs total) (Supplementary Data 2).

**Multiplex capture and amplification.** One hundred nanograms of genomic DNA was hybridized with 1X Ampligase buffer (Epicentre, Madison, WI, USA), 0.32 μM dNTPs, 0.5× of Hemo KlenTaq (0.32 μl; New England Biolabs, Inc., Ipswich, MA, USA), one unit of Ampligase (Epicentre) and MIPs in one 25 μl reaction. Gap filling and ligation were also performed in this reaction. The amount of MIPs needed was based on the 1X pool concentration on a ratio of 800 MIP copies to each haploid genome copy. The reactions were incubated at 95 °C for 10 min and then 60 °C for 22 h. 2 μl of exonuclease mix were used to degrade linear DNA for incubation at 37 °C for 45 min then 95 °C for 2 min. Amplification of the captured DNA was performed as previous reported. Five microlitres of ∼96 different barcoded libraries were pooled and purified with 0.9X AMPure XP beads (Beckman Coulter, Brea, CA, USA) according to standard protocol. Hundred microlitres of 1X EB (Qiagen, Valencia, CA, USA) were used to resuspend the libraries. Then gel visualization with 2% nondenaturing polyacrylamide gel and quantification with the Qubit dsDNA HS Assay (Life Technologies, Grand Island, NY, USA) was performed according to the manufacturer's protocol. Approximately 192 individuals were combined as the final megapools from multiple libraries for sequencing on one lane of Illumina HiSeq 2000 and 101 bp paired-end reads were generated according to the manufacturer's protocol.

**MIP sequencing data analysis.** Sequencing reads were analysed as described previously. For primary single-nucleotide variation, indels and target coverage analysis, initial 101 bp paired-end reads were trimmed to 81 bp before mapping. Sequences were mapped to hg19 using BWA-MEM. Sample-tag indices were made using the 5 bp degenerated sequence, which was removed from the beginning of read2 and added to the index barcode. For QC, read-pairs with incorrect pairs and insert sizes were removed after mapping, leaving only the reads with the highest quality scores. We applied FreeBayes for variant calling, and variants were filtered based on read depth > 8 and read quality > 20 in each megapool. Then all variants were filtered against common variants using dbSNP138. We applied a frequency filter of 'allele count (AC) < 3' (< 0.13%) to the ACGC data set, and annotated variants on the isoform of each gene (Supplementary Data 10) using SeattleSeq138 and the NCBI 37/hg19 reference. The same frequency filters and annotation were used for the ExAC data set (45,376 individuals included, only neuropsychiatric disease-free individuals used). Individuals with < 75% of the target (greater than eightfold coverage) (Supplementary Fig. 1) and genes where less than 30% of the individuals were sufficiently sampled were removed from further analysis (Supplementary Fig. 2).

**Variant validation.** Variants were validated with PCR and Sanger sequencing. Primers were designed using BatchPrimer3 by uploading sequence files in FASTA format according to the user manual with the optimal PCR product size set to 300 bp. For picked primers, we performed BLAT and *in silico* PCR on the UCSC Genome Browser to avoid multiple hits. PCR was performed with a standard programme in 25 μl reaction volume. The PCR products were purified with 0.9X AMPure XP beads and the product was visualized on a gel before sending to Sanger sequencing. To verify the DN status, PCR was also performed on parental DNA, if available.

**Data availability.** The MIP sequencing data for this study can be downloaded from the NIMH data repository National Database for Autism Research (NDAR) at http://dx.doi.org/10.15154/1252218 and is available to all qualified researchers after data use certification. The URLs for data presented herein are as follows: NHLBI Exome Sequencing Project (ESP) Exome Variant Server, http://evs.gs.washington.edu/EVS; UCSC Genome Browser, http://genome.ucsc.edu; MIPgen, https://github.com/shendurelab/MIPGEN; CADD Score, http://cadd.gs.washington.edu; NCBI Gene, http://www.ncbi.nlm.nih.gov/gene; Exome Aggregation Consortium (ExAC), http://exac.broadinstitute.org.

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

## Acknowledgements

This work was supported by grants from the National Basic Research Program of China (2012CB517900) and the National Natural Science Foundation of China (81330027, 81525007 and 31400919). H.G. was supported by the China Postdoctoral Science Foundation (2015M570684), Innovation-Driven Project of Central South University (2016CX038) and Young Talent Lifts Project of CAST. We thank the China Scholarship Council for support for T.W. (201406370028); T.W. was also supported by the Fundamental Research Funds for the Central Universities (2012zzts110). We are grateful to all of the families at the participating Autism Clinical and Genetic Resources in China (ACGC). This research was supported, in part, by the Simons Foundation Autism Research Initiative (SFARI 303241) and NIH (R01MH101221) to E.E.E. H.A.F.S. and T.N.T. were supported, in part, by the NHGRI Interdisciplinary Training in Genome Science Grant (T32HG00035). We thank T. Brown for assistance in editing this manuscript. E.E.E. is an investigator of the Howard Hughes Medical Institute.

## Author contributions

K.X., E.E.E., T.W. and H.G. designed the study; T.W., H.G., B.X., H.A.F.S., H.W., K.H., L.V., W.Z., Y.L. and M.L. performed the experiments; B.P.C. helped with MIP design and data analysis; T.N.T. helped with SSC data analysis; other authors participated in the sample collection and DNA extraction and/or preparation. H.G., T.W., E.E.E., H.A.F.S., B.P.C. and K.X. wrote the manuscript with input from all authors.
