## [Peer Review File · Nature Communications]

Reviewers' comments:

Reviewer #1 (Remarks to the Author):

This manuscript uses an improved version of the molecular inversion probe (MIP) technology to screen for mutations in 189 autism risk genes in a Chinese cohort of over 1,000 trios and 500 ASD individuals. *de novo* and likely gene disrupting (LGD) mutations were identified in 4% of the ASD patients involving 29 genes, with fully 1% of patients showing mutations in the single gene SCN2A.

This paper is an important extension of our knowledge about the genetic basis of ASD in non-European ancestry populations, and is very consistent with observations from other large cohorts based on European ancestry cohorts (e.g., SSC and ASC). The overall frequency of *de novo* LGD mutations was slightly higher in this ACGC cohort, likely due to a significant percentage of patients with intellectual disability (ID). The frequency and pattern of gene mutations did not differ from other large sequencing studies of ID cohorts.

The technical quality of the work in this manuscript is very strong. The ACGC is a consented cohort of patients with confirmed ASD diagnoses intended to be increased over 5 years to a total of 10,000 families (trios/quads) and is a very valuable cohort collection. The MIPS technology is a highly cost-effective and sensitive approach to the screening of sequence variants in a large gene panel such as the one used here.

The manuscript is well written and data is well presented.

Reviewer #2 (Remarks to the Author):

This is an outstanding manuscript. The results are highly informative and useful for all the reasons stated in the manuscript. The methods are excellent. The writing is excellent except for some repetitiveness as stated below.

The findings are not surprising and anything radically different from what was found would generate skepticism, but it is very reassuring to have this beginning towards a truly global dataset of this type. Many of the mutations found help to delineate genes contributing to the etiology of autism.

The only fully significant aspect of the discussion is the following statement: "Interestingly, the rate of DN mutation, however, did not differ significantly from cohorts with ID/DD suggesting that the patients in our Chinese ASD cohort may be enriched for ID/DD." The point might be expanded a bit to comment that any differences between studies are far more likely explained by differences in ascertainment than by differences in genetic contributions to autism. Apart from haplotypes predisposing to *de novo* CNVs there is little evidence of differences in *de novo* mutations among global populations. In fact, the null hypothesis might well be that *de novo* mutations contributing to autism will not differ significantly between global populations. Perhaps global differences in common variants as modifiers or environmental differences could modify penetrance. Global differences in paternal age would also be expected to have effects. If the authors are aware of any evidence for or against ethnic differences in *de novo* mutations, it would be useful to provide a citation and a comment.

There is a moderate amount of repetition of certain information. One example is below. The editor can opine on whether this is excessive.

First statement

The patients with CHD8 DN mutation, for example, show ID (2/2), macrocephaly (2/2), tall stature (2/2), high BMI (2/2), mild regression (2/2), sleep problems (1/2), GI disturbance (1/2), attention deficit (2/2) and anxiety (1/2). These findings are consistent with the clinical and genetic subtype previously reported for this gene²⁵.

Second time in Discussion

We also identified three patients with DN CHD8 LGD mutations. Two of them were recontacted and the phenotypes were similar to those previously reported²⁵ including macrocephaly, tall height, and mild regression.

Minor comment

To our knowledge, we discovered the first DN LGD mutation (insert: in ARHGAP32?) in a patient with ASD although one DN missense variant 12 has been previously reported¹³.

A Beaudet

Reviewer #3 (Remarks to the Author):

Wang et al., report on molecular inversion probe sequencing of 189 risk genes in 1,543 ASD subjects (1,045 trios) with Chinese ancestry.

In terms of its design, this study is very similar in its approach to a paper published in this same journal in 2014, also led by Evan Eichler, which performed MIP resequencing of 64 candidate neurodevelopmental disorder risk genes in 5,979 individuals: 3,486 probands and 2,493 unaffected siblings, which was a follow up to a seminal study in Science from the Eichler / Shendure labs that developed the MIP sequencing technology and applied it to ASD.

The current study confirms the role of CHD8, DSCAM, MECP2, POGZ, & WDFY3 in ASD, and adds weight to genes which previously were only impacted once by de novo mutation GIGYF2, MYT1L, CUL3, DOCK8 and ZNF292.

I have a number of suggestions that I feel would significantly strengthen the paper and a few concerns that require consideration by the authors and editors before this study is suitable for publication.

1. Chinese Ancestry

The authors state in the title that this is a Chinese cohort, and say in the abstract that this study 'highlights the power of global cohorts to assess the impact of the DN LGD risk model'. It's not clear what this means. We expect the pattern of exonic de novo SNVs and indels to be similar across populations, and even if they are different this study is not designed to assess this question.

Furthermore we also expect highly penetrant mutations such as a loss of function mutation in CHD8 will have an ASD/ID phenotype in all populations. What would have been very interesting is if we saw one of the strong candidate genes such as CHD8 had zero mutations in this cohort. Are there any genes that have fewer mutations in this cohort compared to what we expect given studies of ASD in individuals with European ancestry (I think they'll be underpowered to assess this)? Consideration of the population is more important when we study common mutations that have incomplete penetrance and differ in frequency & LD structure between populations, for example in GWAS studies.

I do not think that the individuals are Chinese is a strong selling point for a study of de novo mutation in ASD. Mentioning the fact that the individuals in this study are Chinese in the title and abstract implies that this fact is important when interpreting the genetic results, but they have not shown this

to be the case in the study.

2. ASD Phenotype

The fact that they are Chinese is however important when considering the ASD phenotype, this needs more discussion in the text. In the methods they state that an ASD diagnosis was based on DSM-IV criteria and nothing further. They state that they performed follow up phenotypic characterization of individuals with DNMs in candidate genes. There is no information in the methods about what questions were asked and what measurements were taken in this follow up. They do however mention that almost all children also had ID. This is different to the SSC, where most individuals are high-functioning (IQ>70). As they explain for SCN2A, mutations in also cause Epileptic Encephalopathy & ID, and patients may receive these diagnoses more frequently than ASD in Europe / USA. There have been 21 observations of DN mutations in SCN2A in 4,723 individuals (0.44%) in published studies of ASD (SCN2A DNM n = 14/4,258), ID (SCN2A DNM n = 5/201), and EE (SCN2A DNM n = 2/264). If published studies had proportional numbers of ID and EE cases we'd probably observe a figure close to 1% for SCN2A. So stating that 'DN mutations in SCN2A accounted for a remarkable ~1.1 % of ASD patients' is overselling a result that is neither remarkable nor statistically tested, nor even interesting. Differences in ascertainment of ASD are more likely to explain differences in frequency of DNMs in specific genes. Another alternative they haven't considered is that their methods are more sensitive than previous exome / MIP sequencing studies for identifying de novo mutations in SCN2A (and other genes).

3. Clinical characterization of associated genotypes and phenotypes

Clinical features of some patients are consistent with previous studies. In addition, the authors note for other patients a variety of clinical features that have not been previously described. (P.9, paragraph 3). It's quite unclear whether such features will turn out to be common among subjects with such mutations. Therefore Fig 4B is misleading. This section is purely anecdotal. For example, in a child with a GIGYF2 mutation the authors report the opinion of the mother that "he likes eating food and has no sense of his appetite being satiated". OK, it may be true that this mutation causes this behavior in the child, but they have not provided a statistical argument for this. This anecdotal observation has generated a hypothesis about GIGYF2, but has not tested it. This observation can be mentioned in the discussion but not should be presented as a scientific result. Similarly two of the patients with mutations in different genes walk 'pigeon toed'. OK what does this mean? How pigeon toed are they compared to age matched members of the general public? Is this cohort enriched for pigeon-toed individuals compared to a control cohort?

As more patients come to light with mutations in these genes, it may become possible to make stronger links between these genotypes and phenotypes. For the present study, I would move this entire section to a supplementary information section or table that describes the phenotypes of each child who has a de novo mutation (where available). Figure 4 should be removed entirely; it is not presenting a statistically robust clinical characterization of individuals with mutations in these genes.

4. CADD filtering

The authors select all LGD mutations as well as missense mutations with a CADD score of >30 for further analysis.

I understand they applied this filter, to reduce the total number of missense mutations they had to validate and as a quick and easy way to enrich for causal variants. However, as they go on to say some genes are hit recurrently by de novo missense mutations (e.g. SCN2A and CHD8), and in those instances they go back, remove the filter, and find more de novo variants with CADD scores < 30. These are probably all causative variants as well, CADD is an imperfect measure of pathogenicity, and is not designed to pick out ASD causing mutations. Any de novo missense mutation in an ASD

candidate gene in an individual with ASD should be assumed initially to be pathogenic regardless of the CADD score. Validation of all putative missense DNMs with CADD scores <30 must be performed. All analyses of DNMs should then be repeated without the CADD <30 filter.

5. Analysis of de novo mutations

"Among the 1,045 trios, we discovered 43 DN mutations in 29 of the 189 candidate genes tested, of which 35 were LGD events (Supplementary Table 2). We calculated the overall probability of detecting 35 or more DN LGD events in our panel of 189 genes using a probabilistic model derived from human-chimpanzee differences and an expected rate of 1.75 DN mutations per exome as $p = 1.98 \times 10^{-24}$. This observation corresponds to an odds ratio of 11.1 (95% CI 7.7-15.4) compared to null expectations strongly supporting the enrichment of ASD candidates in our 189 gene set."

OK, but we already know beyond any reasonable doubt that DN mutations in genes like MECP2 and CHD8 and SCN2A etc. cause autism. This analysis should be repeated after removing all these known ASD genes, and instead focus on those that are only hit once in exome studies. In the abstract you should then report the enrichment for this analysis.

"We identify novel DN LGD recurrences (GIGYF2, MYT1L, CUL3, DOCK8 and ZNF292)" What does "establishing recurrence" mean? If this study had sequenced controls instead of cases, you would have also likely have seen recurrence (i.e. a mutation in a gene that has been hit before in a previous study). What would that establish? Are the novel DN LGD recurrences more frequent than would be expected by chance?

The overall rate of DN mutations in the target genes was not significantly different in this cohort compared to European samples, however "When we limited our analysis to only those 25 genes with DN LGD mutations identified in this study, the mutation frequency in Chinese samples was significantly higher than that of European ASD samples ($p = 0.004$, Fisher's exact test, OR = 2.04, 95% CI = 1.22-3.45)". There seems to be an obvious flaw here. The exact set of causal genes that are hit in an individual study are largely based on chance. The results of two different studies will without exception only partially overlap (even if all genes identified are true causal genes). Thus, mutations in study A's genes will always have a lower frequency in study B and vice versa. This does NOT mean that the genetic architectures are different between sample A and sample B.

6. Analysis of inherited variants

There is none as far as I can tell. You could assess the transmission of variants from parents to offspring. Perform a TDT analysis to assess if we see LGD mutations transmitted from parents to the offspring more than we expect by chance? You can use your CADD stratification approach here for missense variants, but apply a range of filters and see if the more pathogenic missense variants get transmitted more frequently than the less pathogenic ones.

7. Are mutations in individual cases de novo or inherited?

"Several LGD mutations in these samples have a high probability of being DN considering the low-predicted mutability and the high rate of DN mutations in these genes in European ASD cohorts."

It's not clear how to interpret the case only LGD mutations.

The claim "High probability of being de novo." Is made without any statistical argument.

LGD variants were observed in 4 genes that have been hit by DN mutation in previous exome studies.

Not much else one can say other than that.

8. Combining MIPs/exome with CNV evidence

The way this is presented in the paper seems somewhat anecdotal to me. There are a couple of examples given of genes (ARHGAP32 and NCOR1) that have been hit by large CNVs and also have individual LGD / missense mutations in this cohort. No formal analysis has been done to combine DNMs and CNVs, I appreciate that this is not a straightforward problem though. I am not convinced yet that ARHGAP32 and NCOR1 are ASD genes based on the evidence presented in Table 2 and Figure 3. Both genes are intersected by a single de novo CNV, both of which are clearly large as they hit ≥ 20 genes. MIPs have not been done for the other 20+ genes so you can't make claims about these genes being important in the CNV locus because of a handful of hits from targeted sequencing. Figure 3 is not clear and is not presenting a robust finding.

Additional minor comments.

1. What do you mean by your "gene model"
2. P. 4 line 5 "More recently, large-scale whole-genome". More recently than what?
3. "It is possible that some of the most severely affected individuals may have been initially recruited from the referring centers. Our patient follow up for specific mutations generally supports this hypothesis. For example, four of the SCN2A families were available for recontact and three of these families had available IQ data that showed impaired cognition for the proband." This statement is lacking much of a basis, what is the evidence that this cohort has lower average IQ? Do other patients with SCN2A mutations have higher IQ?
4. "Although many genes were observed mutated only once in this cohort, a strong case can be made based on the literature that the pathogenic risk variant has likely been discovered." Has been discovered where? Combining this study with previous studies provides strong evidence for which new genes? A formal meta-analysis is needed to support this statement.

Response to reviewer and editorial comments

Reviewer #1 (Remarks to the Author):

This manuscript uses an improved version of the molecular inversion probe (MIP) technology to screen for mutations in 189 autism risk genes in a Chinese cohort of over 1,000 trios and 500 ASD individuals. de novo and likely gene disrupting (LGD) mutations were identified in 4% of the ASD patients involving 29 genes, with fully 1% of patients showing mutations in the single gene SCN2A.

This paper is an important extension of our knowledge about the genetic basis of ASD in non-European ancestry populations, and is very consistent with observations from other large cohorts based on European ancestry cohorts (e.g., SSC and ASC). The overall frequency of de novo LGD mutations was slightly higher in this ACGC cohort, likely due to a significant percentage of patients with intellectual disability (ID). The frequency and pattern of gene mutations did not differ from other large sequencing studies of ID cohorts.

The technical quality of the work in this manuscript is very strong. The ACGC is a consented cohort of patients with confirmed ASD diagnoses intended to be increased over 5 years to a total of 10,000 families (trios/quads) and is a very valuable cohort collection. The MIPS technology is a highly cost-effective and sensitive approach to the screening of sequence variants in a large gene panel such as the one used here.

The manuscript is well written and data is well presented.

We appreciate the referee's kind comments.

Reviewer #2 - A Beaudet (Remarks to the Author):

This is an outstanding manuscript. The results are highly informative and useful for all the reasons stated in the manuscript. The methods are excellent. The writing is excellent except for some repetitiveness as stated below.

The findings are not surprising and anything radically different from what was found would generate skepticism, but it is very reassuring to have this beginning towards a truly global dataset of this type. Many of the mutations found help to delineate genes contributing to the etiology of autism.

The only fully significant aspect of the discussion is the following statement: "Interestingly, the rate of DN mutation, however, did not differ significantly from cohorts with ID/DD suggesting that the patients in our Chinese ASD cohort may be enriched for ID/DD." The point might be expanded a bit to comment that any differences between studies are far more likely explained by differences in ascertainment than by differences in genetic contributions to autism. Apart from haplotypes predisposing to de novo CNVs there is little evidence of differences in de novo mutations among global populations. In fact, the null hypothesis might well be that de novo mutations contributing to autism will not differ significantly between global populations. Perhaps global differences in common variants as modifiers or environmental differences could modify penetrance. Global differences in paternal age would also be expected to have effects. If the authors are aware of any evidence for or against ethnic differences in de novo mutations, it would be useful to provide a citation and a comment.

We agree with the reviewer and have added several sections to the revised manuscript to address possible ascertainment biases that would explain different rates of *de novo* mutations between populations and/or studies. Despite an extensive search of the literature, we found no evidence to suggest ethnic differences regarding *de novo* substitution rates. Indeed, we agree that the null hypothesis should be that there is no difference in *de novo* mutation between ethnic groups. To make this clear, we preface our results with the revised sentence.

“We selected autism risk genes for targeted sequencing mainly based on the frequency and severity of DN mutations from previously published exome sequencing studies^{1,2} under the hypothesis that DN mutations in genes contributing to autism pathology will not differ significantly between global populations.”

Nevertheless, we performed a series of additional analyses, including those suggested. As a surrogate of *de novo* mutation, we tested whether the rates of private variation in our candidate genes differ between ethnic groups. We quantified the distribution of private variation in the 1000 Genomes Project (release: 20130502) by enumerating private variants by gene and population (CEU, CHB, and YRI). We have provided these data in a ternary plot (new Supplementary Figure 4; Supplementary Table 12). Overall, most genes have comparable frequencies of rare variants (CEU:0.3, CHB:0.38, YRI:0.32) supporting the null hypothesis that *de novo* mutations in genes contributing to autism will not differ significantly between global populations. Interestingly, we observed a slight increase in frequency for the CHB (Han Chinese from Beijing); the CHB had the highest proportion of private variants for 95 of the 179 genes we tested possibly consistent with demographic data that this population shows one of the greatest

expansions over the last 5000 years³. In the case of *SCN2A*, however, this factor alone is insufficient to account for the increased rate of *de novo* mutations that we observe in the ACGC compared to other published studies.

We also considered paternal age at the time of conception (Supplementary Table 5). Fathers of individuals who carry a DN LGD or DN MIS30 mutation in the ACGC are not significantly older than those of individuals who do not carry a predicted high-impact DN variant. In fact, ACGC fathers of *de novo* mutation carriers were significantly younger than fathers of *de novo* mutation carriers from published European cohorts⁴ (SSC and The Autism Simplex Collection (TASC) assayed by MIPs; $p = 0.016$, one-tailed Student's *t*-test). Therefore, although increased paternal age at conception has been associated with increased rates of DN mutation, this also does not appear to explain differences in *de novo* mutation rate in the ACGC compared to other studies. While penetrance and environmental modifiers are a possibility, it is far more likely that the increase in DN *SCN2A* mutations is a reflection of ascertainment and possible technological differences (i.e. exome versus molecular inversion probes). Much of our current understanding of European/American autism gene mutation rates has been driven by the SSC (Simons Simplex Collection), which has specifically excluded children with low IQ.

We have updated the text of the supplement to include these new analyses and revised the discussion in the manuscript to comment on the potential effect of ascertainment and technological differences on any differences in the mutation rate.

“We have performed an investigation of DN mutations among ASD candidate risk genes in a Chinese ASD patient cohort. Among the 1,045 trios tested, we discovered 43 DN mutations in 29 of the 189 candidate genes queried by our MIP-based approach. The most frequently DN mutated gene in our study was *SCN2A* where we identified seven novel DN LGD mutations and five DN missense mutations. DN mutations in *SCN2A* accounted for ~1.1 % of ASD patients in our Chinese cohort. This rate was higher than exome- and MIP-sequenced ASD cohorts of European descent. This observation is not likely explained by differences in DN mutation rates among global populations or paternal age but rather by ascertainment and technology biases. For example, published head-to-head comparisons of exome and MIP sequencing technologies on the SSC cohort have shown that MIPs have greater sensitivity for variant detection⁵ which could also account for these increased rates of DN variation in our MIP study compared to exome sequencing studies.

Many sequencing studies of ASD and ID/DD patients have highlighted the extensive overlap among mutated genes between these comorbid conditions^{6,7,8,9}. Indeed, it has been estimated that >40% of patients with ASD may suffer some form of cognitive impairment¹⁰. Even among some of the most rigorously ascertained autism cohorts, such as the SSC, it estimated that 30% of SSC patients also have ID¹¹. Although all of ACGC patients met strict autism (DSM-IV) criteria, IQ data was not routinely collected for the majority of patients in this study. It is possible that some of the most severely affected individuals may have been initially recruited from the referring centers. Our patient follow up for specific mutations generally supports this hypothesis. For example, four of the *SCN2A* families were available for recontact and three of these families had available IQ data that showed impaired cognition for the proband. Further, one of these families also reported severe language impairment adding credence to our hypothesis that this cohort, although defined as autistic by the DSM-IV, is enriched for patients who are

also cognitively impaired¹². This is consistent with studies of autism families in the SSC where three of the original 4 patients identified with *de novo* *SCN2A* mutations were also intellectually impaired (IQ<70)².”

Supplementary Figure 4. The distribution of private variation in the 1000 Genomes Project for the genes targeted by MIPs. The ternary plot compares Northwestern Europeans (CEU; top), Han Chinese from Beijing (CHB; lower right) and Yoruba in Ibadan (YRI; lower left). Each dot, a gene, represents the proportion of private variants contributed by each population; the density of the data is shown as blue contour lines. Dots at the three vertices are genes where private variation is only found in one population. Light blue dots highlight those genes that carried DN LGD mutations in this study (25 genes).

There is a moderate amount of repetition of certain information. One example is below. The editor can opine on whether this is excessive.

*First statement: “The patients with *CHD8* DN mutation, for example, show ID (2/2), macrocephaly (2/2), tall stature (2/2), high BMI (2/2), mild regression (2/2), sleep problems (1/2), GI disturbance (1/2), attention deficit (2/2) and anxiety (1/2). These findings are consistent with the clinical and genetic subtype previously reported for this gene²⁵.”*

*Second time in discussion: “We also identified three patients with DN *CHD8* LGD mutations. Two of them were recontacted and the phenotypes were similar to those previously reported²⁵ including macrocephaly, tall height, and mild regression.”*

We have read through the manuscript and revised sentences to avoid redundancy. For example, we have changed the statement in the discussion to, “We also identified three patients with DN *CHD8* LGD mutations. Two of them were recontacted and the phenotypes were similar to those previously reported²⁵.”

Minor comment: “To our knowledge, we discovered the first DN LGD mutation (insert: in ARHGAP32?) in a patient with ASD although one DN missense variant has been previously reported¹³.”

For clarity, we have inserted “in *ARHGAP32*” as suggested by the reviewer.

Updated text (page 12, paragraph 3): “To our knowledge, we discovered the first DN LGD mutation in *ARHGAP32* in a patient.....”

Reviewer #3 (Remarks to the Author):

Wang et al., report on molecular inversion probe sequencing of 189 risk genes in 1,543 ASD subjects (1,045 trios) with Chinese ancestry. In terms of its design, this study is very similar in its approach to a paper published in this same journal in 2014, also led by Evan Eichler, which performed MIP resequencing of 64 candidate neurodevelopmental disorder risk genes in 5,979 individuals: 3,486 probands and 2,493 unaffected siblings, which was a follow up to a seminal study in Science from the Eichler / Shendure labs that developed the MIP sequencing technology and applied it to ASD.

The current study confirms the role of CHD8, DSCAM, MECP2, POGZ, & WDFY3 in ASD, and adds weight to genes which previously were only impacted once by de novo mutation GIGYF2, MYT1L, CUL3, DOCK8 and ZNF292.

I have a number of suggestions that I feel would significantly strengthen the paper and a few concerns that require consideration by the authors and editors before this study is suitable for publication.

1. Chinese Ancestry

The authors state in the title that this is a Chinese cohort, and say in the abstract that this study 'highlights the power of global cohorts to assess the impact of the DN LGD risk model'. It's not clear what this means. We expect the pattern of exonic de novo SNVs and indels to be similar across populations, and even if they are different this study is not designed to assess this question. Furthermore we also expect highly penetrant mutations such as a loss of function mutation in CHD8 will have an ASD/ID phenotype in all populations. What would have been very interesting is if we saw one of the strong candidate genes such as CHD8 had zero mutations in this cohort. Are there any genes that have fewer mutations in this cohort compared to what we expect given studies of ASD in individuals with European ancestry (I think they'll be underpowered to assess this)? Consideration of the population is more important when we study common mutations that have incomplete penetrance and differ in frequency & LD structure between populations, for example in GWAS studies.

The referee is correct. A two-tailed Fisher's exact test comparing the rate of DN LGD mutations in each of the 189 genes in the Chinese population compared to exome sequencing data from the SSC and the SSC shows that only one gene, *SCN2A*, has a p-value trending toward significance ($p = 9.9 \times 10^{-4}$; OR = 8.0 (95% CI 3.5-Inf). However, this p-value does not survive multiple testing correction (i.e., Bonferroni). A similar analysis of genes depleted of *de novo* variation in the Chinese cohort compared to European cohorts provided no evidence of significance. However, as noted, these analyses likely suffer from a lack of power.

We have revised the text to reflect these findings (page 8, paragraph 1):

“Notably, the rate of *SCN2A* DN LGD mutations appears to be increased in our study (7/1045; 0.7%) compared to published exome sequencing studies of ASD families, specifically the SSC and ASC (4/3,953; 0.1%). This difference was nominally significant ($p = 9.9 \times 10^{-4}$, two-tailed Fisher's exact test, OR = 8.0, 95% CI = 2.0-37.5) but did not withstand multiple testing correction for all 189 candidate

genes. Compared to published MIP-based studies of individuals with ID/DD (3/3387 or 0.09% of individuals carried an *SCN2A* LGD mutation), our data still appear to trend toward increased numbers of individuals carrying DN LGD *SCN2A* mutations ($p = 2.4 \times 10^{-3}$). Because DN LGD mutations in candidate risk genes are individually rare, our current study is underpowered to robustly detect differences in mutation frequencies between affected populations at the single-gene level. However, this nominally significant finding is of interest for potential validation in future studies of larger sample cohorts.”

We have also revised the opening paragraph of the results to make clear the null expectation.

“We selected autism risk genes for targeted sequencing mainly based on the frequency and severity of DN mutations from previously published exome sequencing studies^{1,2} under the hypothesis that DN mutations in genes contributing to autism pathology will not differ significantly between global populations.”

I do not think that the individuals are Chinese is a strong selling point for a study of de novo mutation in ASD. Mentioning the fact that the individuals in this study are Chinese in the title and abstract implies that this fact is important when interpreting the genetic results, but they have not shown this to be the case in the study.

We did not expect the pattern or the types of genes that would be affected to be radically different. In fact, that was the null hypothesis going into the study. The study of these genes in an ethnically distinct population (compared to most of the published data which is from European individuals), however, was important because it strongly suggests that *de novo* mutations are indeed highly penetrant and that genetic modifiers which are likely to be more common and vary in different population groups have little effect. The other selling point for a Chinese “cohort” is the sheer size of the base population (1.38 billion)—given uniform access, it may be critical to access such large numbers of patients to prove pathogenicity of future autism genes. The power of the Chinese cohort, thus, stems from potential future access, the fact that it represents a different genetic background, and that genetic background has negligible effect on *de novo* rates at least for these genes. We, therefore, feel it important to highlight the Chinese origin in both the title and the abstract.

2. ASD Phenotype

*The fact that they are Chinese is however important when considering the ASD phenotype, this needs more discussion in the text. In the methods they state that an ASD diagnosis was based on DSM-IV criteria and nothing further. They state that they performed follow up phenotypic characterization of individuals with DNMs in candidate genes. There is no information in the methods about what questions were asked and what measurements were taken in this follow up. They do however mention that almost all children also had ID. This is different to the SSC, where most individuals are high-functioning ($IQ > 70$). As they explain for *SCN2A*, mutations in also cause Epileptic Encephalopathy & ID, and patients may receive these diagnoses more frequently than ASD in Europe / USA. There have been 21 observations of DN mutations in *SCN2A* in 4,723 individuals (0.44%) in published studies of ASD (*SCN2A* DNM $n = 14/4,258$), ID (*SCN2A* DNM $n = 5/201$), and EE (*SCN2A* DNM $n = 2/264$). If published studies had proportional numbers of ID and EE cases we'd probably observe a figure close to*

1% for SCN2A. So stating that 'DN mutations in SCN2A accounted for a remarkable ~1.1 % of ASD patients' is overselling a result that is neither remarkable nor statistically tested, nor even interesting. Differences in ascertainment of ASD are more likely to explain differences in frequency of DNMs in specific genes. Another alternative they haven't considered is that their methods are more sensitive than previous exome / MIP sequencing studies for identifying de novo mutations in SCN2A (and other genes).

We have now included in the revised manuscript detailed diagnostic criteria from the DSM-IV that we applied (Supplementary Table 3). The recontacted patients underwent a thorough assessment, including a detailed review of medical history and comprehensive physical and neurocognitive phenotyping. Autism Behavior Checklist and SRS data were collected wherever possible. We have updated the Methods section (page 14, paragraph 1):

“Human subjects. All subjects who participated in this study completed informed consent before the original sample collection. DNA samples were extracted from the peripheral blood of patients and parents if available. Proband was diagnosed with ASD based on DSM-IV criteria, and families were excluded if the proband did not meet these diagnostic criteria. Detailed DSM-IV diagnostic criteria are described in Supplementary Table 3. Autism-related single-gene disorders, such as fragile X syndrome, tuberous sclerosis complex and phenylketonuria, were excluded where possible. While patients originate from across China, the majority of samples came from three provinces: namely, Shandong, Hunan and Guangdong (see Supplementary Table 4 for sample distribution). The recontacted patients underwent a thorough assessment, including a detailed review of medical history and comprehensive physical and neurocognitive phenotype. Instruments such as the ABC (autism behavior checklist) and SRS (social responsiveness scale) were applied where possible. This study was approved by the Institutional Review Board of the State Key Laboratory of Medical Genetics, School of Life Sciences at Central South University, Changsha, Hunan, China and adhered to the tenets of the Declaration of Helsinki.”

The reviewer is accurate to point out that ascertainment biases may explain differences in *SCN2A de novo* mutation rates between the Chinese population and exome sequencing studies of European populations. At the request of referee #2, we performed additional analyses such as investigating the effect of paternal age as well as population differences in mutation (see above), but the most likely explanation remains ascertainment bias. We have removed “remarkable” from the manuscript and have revised the text to better describe the biases including the peculiarities of the SSC.

“We have performed an investigation of DN mutations among ASD candidate risk genes in a Chinese ASD patient cohort. Among the 1,045 trios tested, we discovered 43 DN mutations in 29 of the 189 candidate genes queried by our MIP-based approach. The most frequently DN mutated gene in our study was *SCN2A* where we identified seven novel DN LGD mutations and five DN missense mutations. DN mutations in *SCN2A* accounted for ~1.1 % of ASD patients in our Chinese cohort. This rate was higher than exome- and MIP-sequenced ASD cohorts of European descent. This observation is not likely explained by differences in DN mutation rates among global populations or paternal age but rather by ascertainment and technology biases. For example, published head-to-head comparisons of exome and MIP sequencing technologies on the SSC cohort have shown that MIPs have greater sensitivity for variant detection⁵, which could also account for these increased rates of DN variation in our MIP study compared to exome sequencing studies.

Many sequencing studies of ASD and ID/DD patients have highlighted the extensive overlap among mutated genes between these comorbid conditions^{6,7,8,9}. Indeed, it has been estimated that >40% of patients with ASD may suffer some form of cognitive impairment¹⁰. Even among some of the most rigorously ascertained autism cohorts, such as the SSC, it estimated that 30% of SSC patients also have ID¹¹. Although all of AGCG patients met strict autism (DSM-IV) criteria, IQ data was not routinely collected for the majority of patients in this study. It is possible that some of the most severely affected individuals may have been initially recruited from the referring centers. Our patient follow-up for specific mutations generally supports this hypothesis. For example, four of the *SCN2A* families were available for recontact and three of these families had available IQ data that showed impaired cognition for the proband. Further, one of these families also reported severe language impairment adding credence to our hypothesis that this cohort, although defined as autistic by the DSM-IV, is enriched for patients who are also cognitively impaired¹². This is consistent with studies of autism families in the SSC where three of the original four patients identified with DN *SCN2A* mutations were also intellectually impaired (IQ < 70)².”

3. Clinical characterization of associated genotypes and phenotypes

Clinical features of some patients are consistent with previous studies. In addition, the authors note for other patients a variety of clinical features that have not been previously described. (P.9, paragraph 3). It's quite unclear whether such features will turn out to be common among subjects with such mutations. Therefore Fig 4B is misleading. This section is purely anecdotal. For example, in a child with a GIGYF2 mutation the authors report the opinion of the mother that "he likes eating food and has no sense of his appetite being satiated". OK, it may be true that this mutation causes this behavior in the child, but they have not provided a statistical argument for this. This anecdotal observation has generated a hypothesis about GIGYF2, but has not tested it. This observation can be mentioned in the discussion but not should be presented as a scientific result. Similarly two of the patients with mutations in different genes walk 'pigeon toed'. OK what does this mean? How pigeon toed are they compared to age matched members of the general public? Is this cohort enriched for pigeon-toed individuals compared to a control cohort?

As more patients come to light with mutations in these genes, it may become possible to make stronger links between these genotypes and phenotypes. For the present study, I would move this entire section to a supplementary information section or table that describes the phenotypes of each child who has a de novo mutation (where available). Figure 4 should be removed entirely; it is not presenting a statistically robust clinical characterization of individuals with mutations in these genes.

We agree with the referee that each of these clinical observations is, at present, anecdotal, but we feel it is important to flag these for future clinical investigation. We have adopted the suggestion and moved this paragraph from the results to the discussion and have eliminated Figure 4.

Updated text (page 12, paragraph 2):

“We also report here clinical phenotypes for anecdotal cases of DN LGD mutations in *WDFY3*, *GIGYF2*, *MED13L*, *NCKAP1*, *ARHGAP32*, *DSCAM* and *MYTIL*, which serves as a useful starting point for further phenotype–genotype correlations. In addition, we also noted specific comorbidities in several patients that have not been previously described. The patient with a *MYTIL* DN mutation was diagnosed with a motor tic disorder and showed an unusual gait. The patient with a *MED13L* DN mutation has hyperopia and a clear dysmorphism typified by mandibular protrusion. The patient with a *WDFY3* DN LGD mutation has ID and macrocephaly and had severe GI disturbances up to four years of age as well as hyperactivity, sleep problems, attention problems, and anxiety. The patient with a *GIGYF2* DN LGD mutation also has mild ID and macrocephaly. His mother reported an obsession with food but that he has no sense of his appetite being satiated in addition to problems with attention and anxiety.

We note that *MED13L* haploinsufficiency is well established^{13, 14}; others, such as *DSCAM* and *GIGYF2*, are emerging high-impact risk genes from exome and CNV studies^{15, 16, 17}. For example, Akshoomoff and colleagues recently highlighted the neuron-associated GTPase activating protein (*ARHGAP32*) as the best candidate gene for ASD features associated with Jacobsen syndrome based on overlapping deletions that narrowed the critical region to include this locus among three other genes¹⁸. To our knowledge, we discovered the first DN LGD mutation in *ARHGAP32* in a patient with ASD although one DN missense variant has been previously reported¹. Similarly, the discovery of a patient with a DN LGD mutation in *NCOR1* (Table 3), now in addition to published DN missense mutations in ASD² and DD⁹ cohorts, is exciting in light of the known interaction of the Rett syndrome gene (*MECP2*) with members of this nuclear receptor co-repressor complex^{19, 20}.”

4. CADD filtering

The authors select all LGD mutations as well as missense mutations with a CADD score of >30 for further analysis. I understand they applied this filter, to reduce the total number of missense mutations they had to validate and as a quick and easy way to enrich for causal variants. However, as they go on to say some genes are hit recurrently by de novo missense mutations (e.g. SCN2A and CHD8), and in those instances they go back, remove the filter, and find more de novo variants with CADD scores < 30. These are probably all causative variants as well, CADD is an imperfect measure of pathogenicity, and is not designed to pick out ASD causing mutations. Any de novo missense mutation in an ASD candidate gene in an individual with ASD should be assumed initially to be pathogenic regardless of the CADD score. Validation of all putative missense DNMs with CADD scores <30 must be performed. All analyses of DNMs should then be repeated without the CADD <30 filter.

To clarify, *de novo* events were determined after identification of candidate mutations based on frequency and severity. Parents were not run on the MIP platform in the variant discovery phase, only probands. Based on the referee’s request, we reassessed 543 additional missense mutations with lower CADD scores that were private among the 29 genes where a *de novo* mutation was discovered in the ACGC. We were particularly greedy in this approach assessing missense mutations that were also of lower quality. Among these there was insufficient DNA for 21 variants in 21 probands. We tested the 522 available DNA samples and confirmed 429 (or 82.2%) of mutations as validated by Sanger sequencing. We then assessed inheritance in the 386 samples where parental DNA was available (43 samples had one or both parents missing). This analysis identified 11 (2.8%) additional *de novo* mutations (see Supplementary Table 7 of all candidates). This contrasts with our initial selection of *de novo* missense (CADD > 30),

where 21.6% (8/37) validated as *de novo*. While CADD is certainly imperfect, such thresholds clearly increase the odds that a *de novo* and likely pathogenic mutation will be found.

We have added these findings to the main text with additional details in the supplement (Supplementary Figure 3, Supplementary Table 7).

Updated text (page 7, paragraph 1):

“To further refine the total DN mutation rate for our top risk genes, we evaluated all rare missense mutations in the 29 genes where a DN LGD or DN MIS30 event had been identified in the ACGC (Supplementary Table 6). We validated all missense events with a CADD score ≤ 30 (522 events; Supplementary Tables 7 and 8; Supplementary Figure 3 shows the distribution of CADD scores among the 29 genes). From the events that validated in the proband by Sanger sequencing (429/522; 82.2%), we identified 11 (2.8%) new DN missense mutations from samples where parental DNA were available ($n = 386$). This is in contrast to the MIS30 events where 21.6% (8/37) of mutations were DN. While CADD cutoffs are certainly an imperfect definition of pathogenicity, such thresholds significantly increase the odds that a *de novo* and likely disease-causing mutation will be found (CADD > 30 , OR = 8.77, $p = 7.83 \times 10^{-5}$, Fisher’s exact test). Among the low CADD DN missense events, five of the 11 were associated with our top two genes (*SCN2A* and *CHD8*). We identified four additional missense DN mutations (CADD ≤ 30) in *SCN2A*, which increased the total number to 12 DN mutations (7 LGD and 5 missense) (Fig. 2). We also identified one additional DN missense mutation in *CHD8* (for a total of three LGD mutations and one missense mutation) (Fig. 2). DN mutations in *SCN2A*, thus, account for approximately 1.1% of probands in our Chinese cohort.”

Supplemental Figure 3. Increasing CADD scores predict a higher likelihood of a variant arising *de novo*. Shown are the CADD distributions of all missense variants ($n = 480$), detected in ASD patients, from the 29 genes with at least one DN high-impact event (green line) as well as the *de novo* rate at each

CADD threshold (blue bars). Most events (n = 264, 55%) are present in the 10–20 CADD score range; concurrently, we observe very low *de novo* yields in this range (1.9%). At increasing CADD scores, we observe both the expected decrease in overall variant frequencies and a striking increase in the *de novo* fraction at each threshold with a peak at 26.7% of variants with a CADD score over 35. Combining all variants with CADD scores ≥ 30 , we observe 18.6% of all variants as having arisen *de novo*. CADD score thresholds are therefore a useful predictor of likelihood of *de novo* mutation.

Supplementary Table 8. Variant count of the 29 genes with DN mutations identified in ACGC.

Gene	LGD			MIS(CADD>30)			MIS(CADD \leq 30)		
	DN	Validated in trios	Validated in cases	DN	Validated in trios	Validated in cases	DN	Validated in trios	Validated in cases
SCN2A	7	8	8	1	2	3	4	20	20
CHD8	3	3	3	0	1	1	1	20	20
DSCAM	2	3	3	0	2	4	1	18	20
MECP2	2	2	3	0	0	0	0	3	3
ADNP	1	2	2	0	1	1	1	9	11
ARHGAP32	1	1	1	0	0	0	0	20	22
CDKL5	1	3	3	0	0	0	0	3	3
CUL3	1	1	1	0	0	0	0	0	0
DOCK8	1	3	3	0	7	7	0	32	37
DYRK1A	1	1	1	0	1	1	1	7	7
GIGYF2	1	1	2	0	0	1	1	16	16
GRIN2B	1	1	1	0	0	0	0	9	10
MED13L	1	1	1	0	2	2	0	15	18
MYT1L	1	1	1	0	0	0	0	6	7
NCKAP1	1	1	1	0	0	0	0	3	3
NCOR1	1	1	1	0	1	1	0	25	27
PHIP	1	1	1	0	0	0	0	7	7
POGZ	1	1	1	1	3	3	0	13	14
RIMS1	1	5	8	0	1	1	0	14	18
SHANK1	1	1	1	0	0	0	0	10	11
STXBP1	1	1	2	0	1	1	0	2	2
SYNGAP1	1	1	2	0	0	0	0	2	2
TRIP12	1	1	1	0	1	1	0	11	11
WDFY3	1	2	2	1	4	5	1	20	22
ZNF292	1	1	1	0	0	0	0	29	32
ASH1L	0	0	0	2	5	6	1	21	24
CHD2	0	0	0	1	1	1	0	12	14
ITPR1	0	1	1	1	1	1	0	18	21
TSC2	0	0	0	1	3	3	0	21	27
total	35	48	55	8	37	43	11	386	429

5. Analysis of de novo mutations

"Among the 1,045 trios, we discovered 43 DN mutations in 29 of the 189 candidate genes tested, of which 35 were LGD events (Supplementary Table 2). We calculated the overall probability of detecting 35 or more DN LGD events in our panel of 189 genes using a probabilistic model derived from human-chimpanzee differences and an expected rate of 1.75 DN mutations per exome as $p = 1.98 \times 10^{-24}$. This observation corresponds to an odds ratio of 11.1 (95% CI 7.7-15.4) compared to null expectations strongly supporting the enrichment of ASD candidates in our 189 gene set."

OK, but we already know beyond any reasonable doubt that DN mutations in genes like MECP2 and CHD8 and SCN2A etc. cause autism. This analysis should be repeated after removing all these known ASD genes, and instead focus on those that are only hit once in exome studies. In the abstract you should then report the enrichment for this analysis.

We have repeated the analysis removing the known ASD genes and found a significant enrichment (OR = 4.1, $p = 1.17 \times 10^{-5}$). These results have been added to the abstract:

"Recurrent de novo (DN) and likely gene-disruptive (LGD) mutations are important risk factors for autism spectrum disorders (ASD) but have been primarily investigated in cohorts of European ancestry. We sequenced 189 risk genes in 1,543 ASD probands (1,045 from trios) with Chinese ancestry. We report an 11-fold increase in the odds of DN LGD mutations compared to expectation under an exome-wide mutational rate model based on chimpanzee-human divergence. This enrichment for DN LGD mutations remains even after removing known syndromic ASD and intellectual disability genes from our panel ($p = 1.17 \times 10^{-5}$; odds ratio = 4.1). In aggregate, ~4% of ASD patients carry a DN mutation in one of just 29 autism risk genes. The most prevalent gene for recurrent DN mutations was SCN2A (1.1% of patients) followed by CHD8, DSCAM, MECP2, POGZ, WDFY3 and ASHIL. We identify novel DN LGD recurrences (GIGYF2, MYTIL, CUL3, DOCK8 and ZNF292) and DN mutations in genes previously implicated in ASD (ARHGAP32, NCOR1, PHIP, STXBPI, CDKL5 and SHANK1). Patient follow-up confirms phenotypic features associated with the genetic subtypes and highlights how large global cohorts might be leveraged to identify individually rare mutations in genes that together prove pathogenic significance."

"We identify novel DN LGD recurrences (GIGYF2, MYTIL, CUL3, DOCK8 and ZNF292)" What does "establishing recurrence" mean? If this study had sequenced controls instead of cases, you would have also likely have seen recurrence (i.e. a mutation in a gene that has been hit before in a previous study). What would that establish? Are the novel DN LGD recurrences more frequent than would be expected by chance?

We define novel DN LGD recurrence as a second DN LGD mutation in the ACGC in addition to the SSC (Table 2). It was previously determined that the detection of two de novo LGD events in a gene was highly likely to be pathogenic (>95% probability)²¹. Sanders et al. simulated this effect with different sample sizes and showed that even in testing 3,000 samples that the observation of two LGD events was unlikely to occur by chance ($p = 0.01$) (see Figure 2a from Sanders et al., 2012). Moreover, we also consider empirical data for the frequency of observed LGD events in the ExAC database ($n = 45,376$). Of

the five genes highlighted, *MYTIL* shows no evidence of rare LGD mutations, *GIGYF2* and *CUL3* show a few rare LGD mutations (< 0.13%) in this large control population. None of these three genes showed evidence of common LGD variation (see Supplementary Tables 10 and 14). We have clarified this in the main text:

“The identification of two DN LGD mutations in the combined sample size of ACGC and SSC is unlikely²¹. Of these five genes, *MYTIL* shows no evidence of rare LGD mutations in a large population control (ExAC; n=45,376), while two genes had a small number of rare LGD mutations in this control population (*GIGYF2* had nine individuals carrying rare LGD mutations at three sites and *CUL3* had four individuals carrying private LGD variants). None of these three genes showed evidence of common LGD variation in the ExAC cohort (Supplementary Tables 10 and 14)”

The overall rate of DN mutations in the target genes was not significantly different in this cohort compared to European samples, however "When we limited our analysis to only those 25 genes with DN LGD mutations identified in this study, the mutation frequency in Chinese samples was significantly higher than that of European ASD samples (p = 0.004, Fisher's exact test, OR = 2.04, 95% CI = 1.22-3.45)". There seems to be an obvious flaw here. The exact set of causal genes that are hit in an individual study are largely based on chance. The results of two different studies will without exception only partially overlap (even if all genes identified are true causal genes). Thus, mutations in study A's genes will always have a lower frequency in study B and vice versa. This does NOT mean that the genetic architectures are different between sample A and sample B.

The reviewer is correct that this statistical artifact is the result of ascertainment. We have removed the analysis and the sentence.

6. Analysis of inherited variants

There is none as far as I can tell. You could assess the transmission of variants from parents to offspring. Perform a TDT analysis to assess if we see LGD mutations transmitted from parents to the offspring more than we expect by chance? You can use your CADD stratification approach here for missense variants, but apply a range of filters and see if the more pathogenic missense variants get transmitted more frequently than the less pathogenic ones.

Unfortunately, a TDT test is not possible for all the genes because trios were not initially MIP sequenced but rather the probands with candidate mutations were tested for inheritance prospectively. Thus, the presence and transmission of inherited mutations cannot be tested. As mentioned below, this study was primarily focused on *de novo* pathogenic risk variants.

7. Are mutations in individual cases *de novo* or inherited?

*"Several LGD mutations in these samples have a high probability of being DN considering the low-predicted mutability and the high rate of DN mutations in these genes in European ASD cohorts." It's not clear how to interpret the case only LGD mutations. The claim "High probability of being *de novo*." Is*

made without any statistical argument. LGD variants were observed in 4 genes that have been hit by DN mutation in previous exome studies. Not much else one can say other than that.

In this study, we focus primarily on *de novo* mutations. There are, however, 498 autism probands where one or no parental samples were available for testing. Among these, we highlight three potential risk genes (*GIGYF2*, *SYNGAP1* and *STXBPI*) with a higher prior probability of being pathogenic based on our previous analyses of CNV burden²². We formally tested this and now include the data for these three in addition to 38 additional genes that are significant if we integrate available CNV data from patients with developmental delay and autism (see point 8 below). We have clarified this in the text (page 9, paragraph 2) as follows:

“Because there are 498 probands with only one or no parental sample(s) available, inheritance for variants in this subset cannot be determined. Several LGD mutations in these samples have a high probability of being DN considering the low-predicted mutability and evidence of DN mutations in these genes in European ASD cohorts. In addition, our integrated CNV and LGD burden analysis highlight genes with an increased likelihood of haploinsufficiency in ID/DD/ASD that do not reach significance in this cohort by analysis of DN single-nucleotide variants and indels alone. These variants include a frameshift mutation in *GIGYF2* in proband M15067, a stop-gain mutation in *STXBPI* in proband M17663, and a frameshift mutation in *SYNGAP1* in proband M19759. We also identified three LGD mutations in *RIMS1* with undetermined inheritance. Counts of all LGD and missense mutations by gene are provided (Supplementary Table 11).”

8. Combining MIPs/exome with CNV evidence

The way this is presented in the paper seems somewhat anecdotal to me. There are a couple of examples given of genes (ARHGAP32 and NCOR1) that have been hit by large CNVs and also have individual LGD / missense mutations in this cohort. No formal analysis has been done to combine DNMs and CNVs, I appreciate that this is not a straightforward problem though. I am not convinced yet that ARHGAP32 and NCOR1 are ASD genes based on the evidence presented in Table 2 and Figure 3. Both genes are intersected by a single de novo CNV, both of which are clearly large as they hit >=20 genes. MIPs have not been done for the other 20+ genes so you can't make claims about these genes being important in the CNV locus because of a handful of hits from targeted sequencing. Figure 3 is not clear and is not presenting a robust finding.

We performed a more systematic comparison of the 57 autosomal genes with at least one LGD mutation and CNV burden from a recently published set of 29,085 CNV profiles from children with ID/DD and 19,584 CNV profiles from population controls. To control for LGD variants in the general population, we also utilized a filtered version of ExAC with 45,376 individuals without a neuropsychiatric disorder. We agree that this does not provide the same level of confidence regarding a gene's pathogenicity because the CNVs are frequently large. We have added this new analysis and caveats to the main text and supplement (Supplemental Tables 10 and 14).

Updated text (page 9, paragraph 1):

“To further investigate the potential impact of rare inherited LGD variants, we compared our rare LGD event rates with those from the ExAC cohort (using the subset of 45,376 neuropsychiatric disease-free individuals). We combined these data with CNV data from 29,085 children with ID/DD and 19,584 controls using a joint probability model previously described in Coe et al²². Briefly, rates of LGD mutations and deletion CNVs between proband and control cohorts are combined using a statistical model that assumes array and MIP patients are independently assayed. Under this model we examined 57 autosomal genes with at least one LGD mutation and sufficient sequence coverage in both exome and MIP samples. We identified a significant enrichment ($q < 0.05$) for gene-disruptive events for 10 targets (Supplementary Table 10), including three genes with only a single LGD event identified in our study. While these 10 genes are good candidates, most CNVs are large and encompass many genes; in addition, the presence of overlapping large CNVs and DN LGD events does not confer the same specificity as recurrent LGD mutations.”

Additional minor comments.

1. *What do you mean by your "gene model"?*

We have changed “gene model” to “risk gene” throughout.

2. *P. 4 line 5 "More recently, large-scale whole-genome". More recently than what?*

We removed “More recently” because we agree that this is comparative and not relevant.

3. *"It is possible that some of the most severely affected individuals may have been initially recruited from the referring centers. Our patient follow up for specific mutations generally supports this hypothesis. For example, four of the SCN2A families were available for recontact and three of these families had available IQ data that showed impaired cognition for the proband." This statement is lacking much of a basis, what is the evidence that this cohort has lower average IQ? Do other patients with SCN2A mutations have higher IQ?*

Of the six patients carrying *SCN2A* DN mutations in the Simons Simplex Collection (SSC), four have full-scale IQ data available. Three of four patients have $IQ < 70$ and one patient has an $IQ = 114$. In addition, patients with *de novo* mutations in genes that are part of the same co-expression and protein-protein-interaction network were shown to have significantly reduced IQ^{23} . Finally targeted sequencing of *SCN2A* in patients with ID/DD found a significant enrichment LGD burden in this gene²².

We have included some of the basis for this reasoning in the discussion:

“It is possible that some of the most severely affected individuals may have been initially recruited from the referring centers. Our patient follow-up for specific mutations generally supports this hypothesis. For example, four of the *SCN2A* families were available for recontact and three of these families had available IQ data that showed impaired cognition for the proband. Further, one of these families also reported severe language impairment adding credence to our hypothesis that this cohort, although defined as autistic by the DSM-IV, is enriched for patients who are also cognitively impaired¹². This is consistent

with studies of autism families in the SSC where three of the original four patients identified with DN *SCN2A* mutations were also intellectually impaired (IQ < 70)².”

4. "Although many genes were observed mutated only once in this cohort, a strong case can be made based on the literature that the pathogenic risk variant has likely been discovered." Has been discovered where? Combining this study with previous studies provides strong evidence for which new genes? A formal meta-analysis is needed to support this statement.

It has been estimated that 41% of *de novo* LGD mutations are contributing to autism risk². Since many anecdotal reports in the literature may not be pathogenic, we have opted to perform a meta-analysis with single *de novo* LGD and CNV burden (described above) and have eliminated the original sentence.

Other changes made:

1. URLs in the main text have now been removed from the main text and moved to the Web Resources section.

References

1. De Rubeis S, *et al.* Synaptic, transcriptional and chromatin genes disrupted in autism. *Nature* **515**, 209-215 (2014).
2. Iossifov I, *et al.* The contribution of de novo coding mutations to autism spectrum disorder. *Nature* **515**, 216-221 (2014).
3. Genomes Project C, *et al.* A global reference for human genetic variation. *Nature* **526**, 68-74 (2015).
4. O'Roak BJ, *et al.* Recurrent de novo mutations implicate novel genes underlying simplex autism risk. *Nature communications* **5**, 5595 (2014).
5. O'Roak BJ, *et al.* Multiplex targeted sequencing identifies recurrently mutated genes in autism spectrum disorders. *Science* **338**, 1619-1622 (2012).
6. Rauch A, *et al.* Range of genetic mutations associated with severe non-syndromic sporadic intellectual disability: an exome sequencing study. *Lancet* **380**, 1674-1682 (2012).
7. de Ligt J, *et al.* Diagnostic exome sequencing in persons with severe intellectual disability. *The New England journal of medicine* **367**, 1921-1929 (2012).
8. Sanders SJ, *et al.* Insights into Autism Spectrum Disorder Genomic Architecture and Biology from 71 Risk Loci. *Neuron* **87**, 1215-1233 (2015).
9. Deciphering Developmental Disorders S. Large-scale discovery of novel genetic causes of developmental disorders. *Nature* **519**, 223-228 (2015).
10. Charman T, Pickles A, Simonoff E, Chandler S, Loucas T, Baird G. IQ in children with autism spectrum disorders: data from the Special Needs and Autism Project (SNAP). *Psychological medicine* **41**, 619-627 (2011).
11. Fischbach GD, Lord C. The Simons Simplex Collection: a resource for identification of autism genetic risk factors. *Neuron* **68**, 192-195 (2010).
12. Shi X, *et al.* Clinical spectrum of SCN2A mutations. *Brain & development* **34**, 541-545 (2012).
13. Utami KH, *et al.* Impaired development of neural-crest cell-derived organs and intellectual disability caused by MED13L haploinsufficiency. *Hum Mutat* **35**, 1311-1320 (2014).
14. Asadollahi R, *et al.* Dosage changes of MED13L further delineate its role in congenital heart defects and intellectual disability. *Eur J Hum Genet* **21**, 1100-1104 (2013).
15. Turner TN, *et al.* Genome Sequencing of Autism-Affected Families Reveals Disruption of Putative Noncoding Regulatory DNA. *American journal of human genetics* **98**, 58-74 (2016).

16. Gazzellone MJ, *et al.* Copy number variation in Han Chinese individuals with autism spectrum disorder. *Journal of neurodevelopmental disorders* **6**, 34 (2014).
17. Krumm N, *et al.* Excess of rare, inherited truncating mutations in autism. *Nature genetics* **47**, 582-588 (2015).
18. Akshoomoff N, Mattson SN, Grossfeld PD. Evidence for autism spectrum disorder in Jacobsen syndrome: identification of a candidate gene in distal 11q. *Genet Med* **17**, 143-148 (2015).
19. Lyst MJ, *et al.* Rett syndrome mutations abolish the interaction of MeCP2 with the NCoR/SMRT co-repressor. *Nature neuroscience* **16**, 898-902 (2013).
20. Ebert DH, *et al.* Activity-dependent phosphorylation of MeCP2 threonine 308 regulates interaction with NCoR. *Nature* **499**, 341-345 (2013).
21. Sanders SJ, *et al.* De novo mutations revealed by whole-exome sequencing are strongly associated with autism. *Nature* **485**, 237-241 (2012).
22. Coe BP, *et al.* Refining analyses of copy number variation identifies specific genes associated with developmental delay. *Nature genetics* **46**, 1063-1071 (2014).
23. Hormozdiari F, Penn O, Borenstein E, Eichler EE. The discovery of integrated gene networks for autism and related disorders. *Genome research* **25**, 142-154 (2015).

REVIEWERS' COMMENTS:

Reviewer #2 (Remarks to the Author):

Strongly support publication.

Reviewer #3 (Remarks to the Author):

The authors have adequately addressed my concerns